

# Optimal model complexity for terrestrial carbon cycle prediction

Caroline A. Famiglietti[1,*], T. Luke Smallman[2], Paul A. Levine[3], Sophie Flack-Prain[2], Gregory R. Quetin[1], Victoria Meyer[4], Nicholas C. Parazoo[3], Stephanie G. Stettz[3], Yan Yang[3], Damien Bonal[5], A. Anthony Bloom[3], Mathew Williams[2], and Alexandra G. Konings[1]

[1]Department of Earth System Science, Stanford University, Stanford, USA
[2]School of GeoSciences and National Centre for Earth Observation, University of Edinburgh, Edinburgh, UK
[3]Jet Propulsion Laboratory, California Institute of Technology, Pasadena, USA
[4]School of the Art Institute of Chicago, Chicago, USA
[5]Université de Lorraine, AgroParisTech, INRAE, UMR Silva, 54000 Nancy, France

*Correspondence to*: Caroline A. Famiglietti (cfamigli@stanford.edu)

**Abstract.** The terrestrial carbon cycle plays a critical role in modulating the interactions of climate with the Earth system, but different models often make vastly different predictions of its behavior. Efforts to reduce model uncertainty have commonly focused on model structure, namely by introducing additional processes and increasing structural complexity. However, the extent to which increased structural complexity can directly improve predictive skill is unclear. While adding processes may improve realism, the resulting models are often encumbered by a greater number of poorly-determined or over-generalized parameters. To guide efficient model development, here we map the theoretical relationship between model complexity and predictive skill. To do so, we developed 16 structurally distinct carbon cycle models spanning an axis of complexity and incorporated them into a model–data fusion system. We calibrated each model at 6 globally-distributed eddy covariance sites with long observation time series and under 42 data scenarios that resulted in different degrees of parameter uncertainty. For each combination of site, data scenario, and model, we then predicted net ecosystem exchange (NEE) and leaf area index (LAI) for validation against independent local site data. Though the maximum model complexity we evaluated is lower than most traditional terrestrial biosphere models, the complexity range we explored provides universal insight into the inter-relationship between structural uncertainty, parametric uncertainty, and model forecast skill. Specifically, increased complexity only improves forecast skill if parameters are adequately informed (*e.g.,* when NEE observations are used for calibration). Otherwise, increased complexity can degrade skill and an intermediate-complexity model is optimal. This finding remains consistent regardless of whether NEE or LAI is predicted. Our COMPLexity EXperiment (COMPLEX) highlights the importance of robust, observation-based parameterization for land surface modeling and suggests that data characterizing net carbon fluxes will be key to improving decadal predictions of high-dimensional terrestrial biosphere models.

## 1 Introduction

The role of the terrestrial biosphere in the global carbon cycle is challenging to model *(Friedlingstein et al., 2013)* due to the diverse processes, forcings, and feedbacks driving variability of gross fluxes *(Heimann & Reichstein, 2008; Luo et al., 2015).*



Many attempts to reduce model uncertainty have focused on matching models to nature by representing an increasing number of processes known to influence different parts of the carbon cycle (*e.g.,* vegetation demography *[R. A. Fisher et al., 2018]* or
plant hydraulics *[Kennedy et al., 2019])*. In this way, models of the terrestrial biosphere have become more complex over time *(J. B. Fisher et al., 2014; Bonan, 2019; R. A. Fisher & Koven, 2020)*. Despite such advancements, the spread in terrestrial carbon cycle predictions remains large *(Arora et al., 2020)* and is dominated more so by model uncertainty than by either internal variability of the climate system or emission scenario uncertainty *(Lovenduski & Bonan, 2017; Bonan & Doney, 2018)*. Because the behavior of the terrestrial biosphere feeds back directly on the rate of CO2 accumulation in the atmosphere,
understanding the most effective ways of reducing this model uncertainty is crucial. Progress can benefit not only long-term predictions of global change, but also near-term, regional-scale ecological forecasts aimed to inform sustainable decision-making *(Dietze et al., 2018; Thomas et al., 2018; White et al., 2019)* and modeling studies focused on understanding the recent past *(Schwalm et al., 2020)*.

While ecological models are becoming more and more detailed, the extent to which predictive skill scales with model
complexity is not clear. The logic behind enhancing model realism with increased complexity is intuitive: a highly simplistic model may be structurally unable to capture key relationships defining the system (it underfits), which would naturally imply that greater detail is needed to improve model performance. However, excessively complex models have their own limitations. Because they often contain more parameters than can be robustly determined with the available data *(e.g., Prentice et al., 2015; Shi et al., 2018; Feng, 2020)*, they are prone to learning "noise" instead of true interactions (also called overfitting;
*Ginzburg & Jensen, 2004; Hawkins, 2004; Keenan et al., 2013)*. Equifinality—the case in which vastly different parameter sets can yield similar model performance *(Beven, 1993; Beven & Freer, 2001)*—also becomes more likely as model complexity increases. This dichotomy between model complexity and model performance is known in the statistics and machine learning communities as the bias–variance tradeoff. According to this theory, a model that balances the costs of under- and overfitting can minimize forecast error *(Lever et al., 2016)*. It is therefore possible that other approaches to reducing carbon cycle model
uncertainty (*e.g.,* improving model parameterization) may be more effective than increasing structural realism in some circumstances, as also noted by *Shiklomanov et al., 2020* and *Wu et al., 2020a*.

Here, we explicitly map the relationship between model complexity and predictive performance across a spectrum of model structures and parameterizations, hypothesizing that an intermediate-complexity carbon cycle model can outperform a low- or high-complexity one. Our approach can inform ecological models that operate on a spectrum of scales, from localized
at the level of individual stands to highly generalizable across the global land surface. This study is particularly relevant for global ecological models, which often function as the land surface component of large-scale Earth system models and have been employed in contexts that carry significant policy relevance (*e.g.*, Intergovernmental Panel on Climate Change [IPCC] reports; *Stocker et al., 2014)*. Hereafter we refer to such models as TBMs, or terrestrial biosphere models.

We note a distinction between conceptualizing complexity as a straightforward count of a model's parameters, equations,
or processes, versus as an emergent property of its solution space. When locations or data constraints do not allow certain model parameter values or modeled states, this reduces the effective complexity of the remaining set of possible solutions.



That is, one can consider what we term the "effective complexity" of a model as a function of the actual parameter combinations that are possible for that model, or equivalently, the volume of space occupied by these parameter combinations. Two models with the same number of parameters may have very different effective complexities, for example, because correlations between

parameters (*e.g.,* allocation fraction to foliage and turnover rate of foliage *[Fox et al., 2009])* or the extent to which they are constrained (*i.e.*, many more states are possible in the absence of assimilated data than in the presence of it *[Keenan et al., 2013]*, or when the assimilated data has high uncertainty) can influence the models' effective degrees of freedom. As a simple analogy, consider the difference between a sphere and a disc in three-dimensional space *(Fig. 1)*. Although both exist within the space determined by 3 unconstrained parameters (axes), they are not identical because the *volumes* they occupy—and the

relationships between their parameters—are drastically different. The same can be true between models: one model's equations or assimilated observations may constrain the dimensionality of its potential parameter space to "resemble" a disc, while that occupied by another, less constrained model may look more like a sphere.

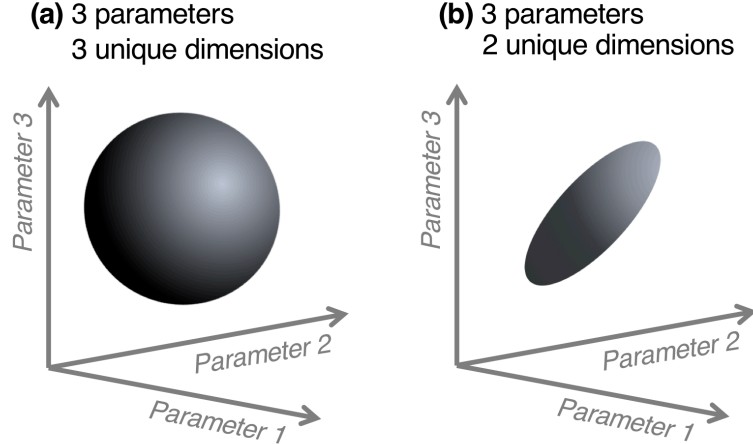

**Figure 1:** Conceptual diagram of effective complexity in 3-parameter space. A sphere (a) has three unique dimensions spanning the three axes of variability (analogous to a larger solution space for a given model). In the region defined by the same three axes, a disc (b) has only two unique dimensions (analogous to a smaller solution space, perhaps due to two parameters being highly correlated).

Model–data fusion (MDF) systems (also known as data assimilation systems) provide an effective way of isolating and evaluating different model structures by using observations to derive optimized model parameters with uncertainty. An increasingly common tool for carbon cycle science, MDF has been leveraged to provide insight into long-term trends of carbon fluxes *(e.g., Rayner et al., 2005),* to reconcile the roles of specific datasets in constraining parametric uncertainty *(e.g., Keenan et al., 2013)*, and more *(Scholze et al., 2017)*. Here we use an MDF system called the CARbon DAta MOdel fraMework, or

CARDAMOM *(Bloom & Williams, 2015; Bloom et al., 2016)*, chosen because of its high customizability. The structure of its underlying ecosystem carbon model, DALEC *(Williams et al., 2005; Bloom & Williams, 2015)*, can be easily adjusted to



become more simple or detailed (*e.g.,* by changing the number of carbon pools or by modifying the functional representations of certain carbon fluxes). Various combinations of observational and functional constraints can also be tested in the assimilation process, along with different assumptions on the amount of error inherent to each assimilated dataset (the

characterization of which is an ongoing challenge for the modeling community *[Keenan et al., 2011])*. Taken together, this flexibility allows for experimentation with the different levers that control effective model complexity.

In this paper, we demonstrate the extent to which the prediction accuracy of two key carbon cycle variables can theoretically scale with model complexity. Net ecosystem exchange (NEE) and leaf area index (LAI) were chosen for the analysis because they represent integrated effects of different parts of the carbon cycle (NEE is the balance of photosynthesis

and ecosystem respiration fluxes, while LAI strongly controls canopy photosynthesis *[Bonan, 1993])*. Additionally, both are commonly measured and modeled. To explore the complexity–skill relationship, we developed 16 structurally distinct carbon cycle models *(i.e.,* variants of the DALEC model) spanning a range of complexity and calibrated them using the CARDAMOM framework. Several recent studies have demonstrated the utility of CARDAMOM for understanding multiple aspects of the carbon cycle *(e.g., Konings et al., 2019; López-Blanco et al., 2019; Bloom et al., 2020; Quetin et al., 2020; Yin et al., 2020)*,

lending confidence for its use here. We calibrated each DALEC variant within CARDAMOM under 42 different data scenarios *(i.e.,* combinations of data constraints and assumptions about observational error) representing different degrees of certainty with which parameters are determined. Each model was calibrated and validated at 6 globally-distributed eddy covariance sites covering a range of biomes and vegetation types, with data collected over multiple years. To quantify complexity, we computed the effective complexity of each model calibration using a principal component analysis (PCA) that reduced the parameter

space to its primary axes of variance. Forecast skill was determined using an overlap metric that takes account of uncertainty both in the model forecast and the validation data. Though the range of complexity we evaluated here is lower than that populated by large-scale TBMs, this experiment reveals universal modeling elements that control performance. Specifically, here our COMPLexity EXperiment (COMPLEX) aims to answer the following questions: (a) What controls a given model run's effective complexity? (b) Under what conditions does increasing model complexity improve forecast skill?

## 2 Methods

### 2.1 Suite of carbon cycle models (DALEC variants)

The Data Assimilation Linked Ecosystem Carbon (DALEC) model suite includes 16 related intermediate-complexity models of the terrestrial carbon cycle. Each model variant tracks the state and dynamics of both live and dead carbon pools, their interactions, and responses to meteorology and disturbance such as fire or biomass removals. From an initial DALEC model

*(Williams et al., 2005)*, we produced alternate structures that either aimed to reduce complexity by focusing on core variables/processes and removing others, or aimed to increase complexity by including hypothesized missing carbon pools or improving on over-simplified processes.





Accordingly, the DALEC suite spans a range of model structures (*i.e.,* number of carbon pools, carbon pool connectivity) and process representations (component sub-models of varying complexity) related to different simulations of photosynthesis,

plant respiration, decomposition, and water cycle feedbacks. These representations are listed in *Table 1* and described in further detail in *Appendix A*. To facilitate disentanglement of the impacts of specific alternate process representations, the different sub-models can be related to a common baseline structure of the carbon cycle (*Fig. 2a*). Specific variants of this general structure for the least and most detailed models in this analysis are presented in *Fig. 2b-c*, while additional diagrams for the remaining models are shown in *Appendix B (Fig. B1-7)*. Across models, carbon enters the system via gross primary

productivity (GPP), which is allocated to autotrophic respiration ($R_a$) and non-canopy live tissues based on fixed fractions. Canopy growth and mortality is determined by a phenology sub-model which is sensitive either to day of year (sub-model scheme CDEA), environmental factors (GSI) or a combination of environmental factors and estimated net canopy carbon export (NCCE). Mortality of wood and fine roots follows continuous turnover based on first order kinetics. Decomposition of dead organic matter and associated heterotrophic respiration ($R_h$) follows first order kinetics with an exponential temperature

sensitivity (and, in models C2-C5, a linear soil moisture sensitivity).

**Table 1:** Summary of the DALEC sub-model combinations assessed in COMPLEX. For detailed description see supporting material. ID is model identifier. CDEA = Combined Deciduous Evergreen Analytical model, CDEA+ = CDEA with variable labile release fraction, GSI = Growing Season Index, NCCE = Net Canopy Carbon Export, ACM = Aggregated Canopy Model,

T = temperature, M = soil moisture, CUE = carbon use efficiency. fNPP:GPP indicates a fixed fractional allocation of gross primary production (GPP) to foliage net primary production (NPP). DOM is dead organic matter.

| ID | Canopy phenology | Method of computing GPP | Water cycle | $R_h$ | CUE | Number of parameters | DOM pools | Live pools |
|---|---|---|---|---|---|---|---|---|
| C1 | CDEA | ACM v1 | No | T | $R_a$:GPP | 23 | 2 | 4 |
| C2 | CDEA+ | ACM v1 | Yes | T+M | $R_a$:GPP | 33 | 2 | 4 |
| C3[*] | CDEA+ | ACM v1 | Yes | T+M | $R_a$:GPP | 35 | 2 | 4 |
| C4[†] | CDEA+ | ACM v1 | Yes | T+M | $R_a$:GPP | 34 | 2 | 4 |
| C5 | CDEA+ | Analytical Ball-Berry | Yes | T+M | $R_a$:GPP | 34 | 2 | 4 |
| C6 | CDEA | ACM v2 | No | T | $R_a$:GPP | 23 | 2 | 4 |
| C7 | CDEA | ACM v2 | Yes | T | $R_a$:GPP | 27 | 2 | 4 |
| C8[‡] | CDEA | ACM v1 | Yes | T | $R_a$:GPP | 36 | 2 | 4 |





| | | | | | | | | |
|---|---|---|---|---|---|---|---|---|
| E1 | fNPP:GPP | ACM v1 | No | T | $R_a$:GPP | 17 | 3 | 3 |
| G1 | GSI | ACM v2 | No | T | $R_m$:GPP + $R_g$:NPP | 37 | 3 | 4 |
| G2 | GSI | ACM v2 | Yes | T | $R_m$:GPP + $R_g$:NPP | 40 | 3 | 4 |
| G3 | GSI + NCCE | ACM v2 | No | T | $R_m$Leaf(T) + $R_m$Wood:GPP + $R_m$Root:GPP + $R_g$:NPP | 43 | 3 | 4 |
| G4 | GSI + NCCE | ACM v2 | Yes | T | $R_m$Leaf(T) + $R_m$Wood:GPP + $R_m$Root:GPP + $R_g$:NPP | 43 | 3 | 4 |
| S1 | fNPP:GPP | ACM v1 | No | T | $R_a$:GPP | 11 | 1 | 2 |
| S2 | CDEA | ACM v1 | No | T | $R_a$:GPP | 14 | 1 | 3 |
| S4 | CDEA | ACM v1 | No | T | $R_a$:GPP | 17 | 3 | 2 |

*Includes cold weather GPP limitation

†Includes surface runoff parameterization (assumes constant runoff to infiltration ratio at surface)

‡Includes two water storage pools (plant-available and plant-unavailable water)



**(a) General DALEC structure**

**(b) Simplest model (S1)**

**(c) Most complex model (G1-G4)**

**Figure 2:** Overview of the carbon pools (filled boxes) and fluxes (arrows, with names in open boxes) represented in the DALEC model suite. (a) Broad structure of the DALEC model maintained across all variants in the suite; (b) carbon cycle structure of the simplest model; (c) carbon cycle structure of the most detailed model.

## 2.2 Site selection

The COMPLEX experiment uses information from 6 globally-distributed eddy covariance sites participating in FLUXNET *(Pastorello et al., 2020)* (Table 2). Our site selection procedure aimed to maximize biogeographical spread and diversity of natural ecosystems while fulfilling specific data requirements. These constraints collectively yielded a series of site selection criteria that are described in detail in *Appendix C*. As an example, the sites must not be dominated by the C4 photosynthetic pathway, nor arable agriculture nor intensively grazed grassland. Additionally, we required that the range of time series





observations to be used for model calibration and validation spanned at least a decade. Data collated at each site are described
below (see *Sect. 2.3)*.


**Table 2:** Summary of sites, showing their location, FLUXNET code, observational time period, mean climate information and
ecosystem type. Latitude is given in -90/90 and longitude is -180/180. Ecosystem type is denoted using the International
Geosphere-Biosphere Programme (IGBP) classification. DBF = deciduous broadleaf forest; EBF = evergreen broadleaf forest;
ENF = evergreen needleleaf forest; WSA = woody savanna.

| Site Name | Site Code | Reference | Latitude | Longitude | IGBP | Data record | Mean annual temp. [°C] | Mean annual precip. [mm/yr] |
|---|---|---|---|---|---|---|---|---|
| Howard Springs | AU-How | *Beringer et al., 2007* | -12.4943 | 131.1523 | WSA | 2001-2014 | 27.0 | 1449 |
| Hyytiala | FI-Hyy | *Suni et al., 2003* | 61.84741 | 24.29477 | ENF | 1999-2014 | 3.8 | 709 |
| Le Bray | FR-LBr | *Berbigier et al., 2001* | 44.71711 | -0.7693 | ENF | 1998-2008 | 13.6 | 900 |
| Puechabon | FR-Pue | *Rambal et al., 2004* | 43.7413 | 3.5957 | EBF | 2000-2014 | 13.5 | 883 |
| Guyaflux | GF-Guy | *Aguilos et al., 2018* | 5.27877 | -52.92486 | EBF | 2004-2018 | 25.7 | 3041 |
| Harvard Forest | US-Ha1 | *Munger & Wofsy, 2020a, 2020b* | 42.5378 | -72.1715 | DBF | 1998-2012 | 6.2 | 1071 |


### 2.3 Model–data fusion

We used the CARDAMOM model–data fusion system *(Bloom & Williams, 2015; Bloom et al., 2016)* to parameterize the
DALEC model suite with available observations of the carbon cycle. Specifically, we employed Bayesian inference to retrieve
time-invariant, site-specific, optimized parameters and initial conditions for a given DALEC model (y) as informed by

observations (O), where $p(y|O) \propto p(y) \cdot p(O|y)$. Here, $p(y|O)$ is the posterior parameter probability distribution, $p(y)$ is the
prior parameter probability distribution, and $p(O|y)$ is proportional to the likelihood of parameters y given observations O.

For each model, $p(y)$ is derived as the product of *(i)* the prior probability density functions for each model parameter, and
*(ii)* ecological and dynamical constraints (EDCs; *i.e.,* functional constraints). EDCs are simple mathematical functions that
impose conditions on inter-relationships between model parameters based on known ecological theory. They are used to inform



parameter prior information with broader ecological knowledge and tend to reduce bias and equifinality *(Bloom & Williams, 2015)*. One example of an EDC in CARDAMOM is the imposed constraint that litter turnover times are faster than soil organic matter turnover times *(e.g., Gaudinski et al., 2000)*. In this analysis, each model includes some or all of the EDCs documented in *Bloom et al. (2016)*.

   The likelihood $p(O|y)$ is derived as a function of the mismatch between observations O and the model realization M

corresponding to y, such that $p(O|y) \propto exp\left(-\frac{1}{2}\sum_{n=1}^{N}\left(\frac{O_n - M_n}{\sigma_n}\right)^2\right)$, where $\sigma_n$ is the error for the *n*th observation. This formulation requires no assumptions on the normality of prior or posterior parameter distributions and is robust to missing data. In our analysis, monthly-averaged eddy covariance NEE measurements from FLUXNET, monthly-averaged leaf area index (LAI) estimates from the Copernicus Global Land Service *(Verger et al., 2014; Fuster et al., 2020)*, and in situ wood stock surveys were made available for ingestion into the model (see *Appendix C*). NEE uncertainty was assumed to be 0.58

gC m$^{-2}$ day$^{-1}$ based on estimates of random errors in eddy covariance measurements from *Hill et al. (2012)*. A time-varying uncertainty estimate was included with the Copernicus LAI product and site-specific, locally-derived biomass uncertainties were provided by the site PI or drawn from relevant publications when necessary. Model drivers included monthly average site meteorology (air temperature, shortwave radiation, atmospheric $CO_2$ concentration, vapor pressure deficit, precipitation and wind speed). Here models were run at the monthly timestep.

To sample the distribution $p(y|O)$ (namely the product of $p(O|y)$ and $p(y)$), we used an adaptive proposal Metropolis-Hastings Markov Chain Monte Carlo (MCMC) approach *(Haario et al., 2001)*. We performed $10^8$ iterations for each of four chains, which were checked for convergence using the Gelman-Rubin criterion (<1.2). A subset of 100 samples of *y* was selected from the latter half of each chain for our analysis. For additional details on the implementation of this algorithm within CARDAMOM, see *Bloom & Williams (2015)*.

**2.4 Experimental design**

   We performed a factorial experiment such that each of the 16 structurally distinct carbon cycle models was run within CARDAMOM under all possible combinations of sites, observational and functional constraints, and assumptions on data uncertainties. These scenarios represented differing degrees of certainty with which parameter distributions were determined. Specifically, we considered *(a)* 6 sites; *(b)* 6 options for assimilated data, including one for which no data was ingested into

the model; *(c)* 4 options for the magnitude of error assumed on the assimilated datasets (represented by scalar multipliers on the prescribed nominal uncertainties); and *(d)* 2 options for EDC state (either present or absent) *(Table 3)*. In total, this factorial approach yielded 4032 unique model runs (16 models × 6 sites × 21 data scenarios × 2 EDC states). Using a high number of factorial model runs both added robustness to our interpretation and allowed for consideration of each factor's influence across a range of background conditions.





**Table 3:** Model specifications varied in the factorial experiment. Each of the 16 model versions was run with every combination of scenarios across each variable. Note that observational error scalars were not applied when no data were assimilated into the model.

| Variable | Scenarios |
|---:|:---|
| *Site* | AU-How<br>FI-Hyy<br>FR-LBr<br>FR-Pue<br>GF-Guy<br>US-Ha1 |
| *Assimilated data* | NEE<br>NEE, LAI<br>NEE, LAI, biomass<br>LAI<br>LAI, biomass<br>None |
| *Observational error scalar* | 50%<br>100%<br>150%<br>200% |
| *EDC state* | All present<br>All absent |

*Fig. 3* shows examples of three model analyses at the FR-LBr site, highlighting the range in NEE prediction performance across different model structures and data scenarios. Each model run contains a calibration period (the first 5 years of the site record; shown in white) during which optimized parameters were derived, and a forecast period (the remaining years of the record, which always spanned at least 5 years because no site contained fewer than 10 years of data; shown in gray) during which fluxes and pools were predicted with the optimally parameterized model. In the scenario presented, model S2 is highly

constrained by multiple datasets *(Fig. 3a)*. By contrast, model C2 is moderately constrained *(Fig. 3b)* and model G4 is poorly constrained *(Fig. 3c)*, which is evident by comparing the relative uncertainty of the NEE forecasts (blue shading) for each model. Accounting for prediction uncertainty—as well as data uncertainty (red shading)—is a key goal of our model skill evaluation approach. Forecast skill for each model run was computed by comparing predictions and observations drawn strictly from the forecast period, using the histogram intersection algorithm (see *Sect. 2.5.1*). The complexity of each run was

quantified based on its effective complexity (see *Sect. 2.5.2*).



**Figure 3:** Example model runs (title of each subplot) at the FR-LBr site. The calibration window—the first 5 years of the record—is shown in white and the forecast window is shaded gray. The ensemble spread (blue shading) encapsulates the 5th–95th percentile of runs. (a) Forecast skill = 0.09; Effective complexity = 4; (b) Forecast skill = 0.34; Effective complexity = 24; (c) Forecast skill = 0.21; Effective complexity = 39.



## 2.5 Analysis

### 2.5.1 Skill metric

We chose the histogram intersection as a skill metric because it captures accuracy along with both prediction uncertainty (*i.e.,* the ensemble spread for a given model output) and observational uncertainty (*i.e.,* the mean value and error for a given observation). This approach contrasts with more familiar metrics such as the coefficient of determination ($R^2$) or root-mean-square error (RMSE), which do not account for uncertainties surrounding individual data points or predictions.

The histogram intersection is a simple algorithm that calculates the similarity of two discretized probability distributions $p$ and $q$ and is commonly used in the machine learning community *(e.g.,* for image classification; *Jia et al., 2006; Maji et al.,*
*2008)*. Specifically, the histogram intersection of $p$ and $q$ is computed as $\sum_{i=1}^{n} min(p_i, q_i)$ where $n$ is the number of bins in the two histograms. In our case, $p$ was the predicted NEE or LAI ensemble for a given timestep and $q$ was a Gaussian distribution with mean and standard deviation equivalent to the observed NEE or LAI value and its error, respectively. The metric is bounded between 0 (no overlap) and 1 (identical distributions). Because histograms $p$ and $q$ correspond to individual months in the forecast period, the metric used for analysis was the average histogram intersection over all such months.

We note that results for NEE predictions are presented in the main figures of this paper, while those for LAI predictions are included in the supporting information.

### 2.5.2 Complexity metric

The effective complexity of each model run was computed using a principal component analysis (PCA) on the posterior parameter space. When applied to CARDAMOM output, the PCA reduces the posterior parameter space (*n* ensembles of *m*
parameters) to a set of at most *m* uncorrelated variables that successively maximize variance. As such, this approach finds the smallest number of unique dimensions necessary to explain the most variability in the posterior parameter space of each model analysis. Specifically, we defined effective complexity as the number of principal components for which 95% of variance in the posterior parameter space was explained. Note that in our experiment, a given DALEC model variant has a distribution of effective complexities corresponding to the different specifications for each run (*i.e.,* data scenario, site; *Table 3*).

## 3 Results

### 3.1 Behavior of effective complexity metric

Effective complexity—defined as the number of principal components for which 95% of the variance in the posterior parameter space is explained (see *Sect. 2.5.2*)—is primarily determined by model structure *(Fig. 4a, inset)*. Specifically, over all runs





included in the experiment, effective complexity varies far more between different models than between the other tested factors
(assimilated data, observational error scalar, site, and EDC presence/absence). This link to model structure provides insight
into the metric's interpretability and justifies its use as a measure of model complexity.

While predominantly determined by the choice of model, effective complexity also varies according to the degree to which
parameters are constrained *(Fig. 4a)*. It therefore captures the inter-relationship between model structure and parameterization.
Within a given model structure, each of the experimentally varied factors yields a range of distinct complexities that follows
a predictable pattern: effective complexity is higher for runs with weaker constraints on parameters than it is for runs with
stronger constraints on parameters. This is easily interpretable in the case of assimilated data, which is the dominant within-
model control on effective complexity *(Fig. 4b)*. Runs for which no observations are ingested into the model have consistently
higher effective complexities than runs for which NEE, LAI, and biomass observations are all ingested (compare yellow and
purple circles in *Fig. 4b)*, since the observational constraints reduce the possible model solution space. Similar behavior is also
observed across the different error scalars tested in the experiment (larger observational error assumptions correspond to higher
effective complexities *[Fig. S1]*) and between the presence versus absence of EDCs (the absence of non-observational realism
constraints yields higher effective complexities *[Fig. S2]*). Conceptually, this pattern can be understood in the following way.
Parameters in a given model's high-complexity runs were sampled from wider posterior distributions (due to weak or absent
constraints) than in its low-complexity runs. This implies greater variance between parameter sets selected in high-complexity
runs—and thus more distinct dimensions of variability in the posterior parameter space—than in low-complexity runs for the
same model.

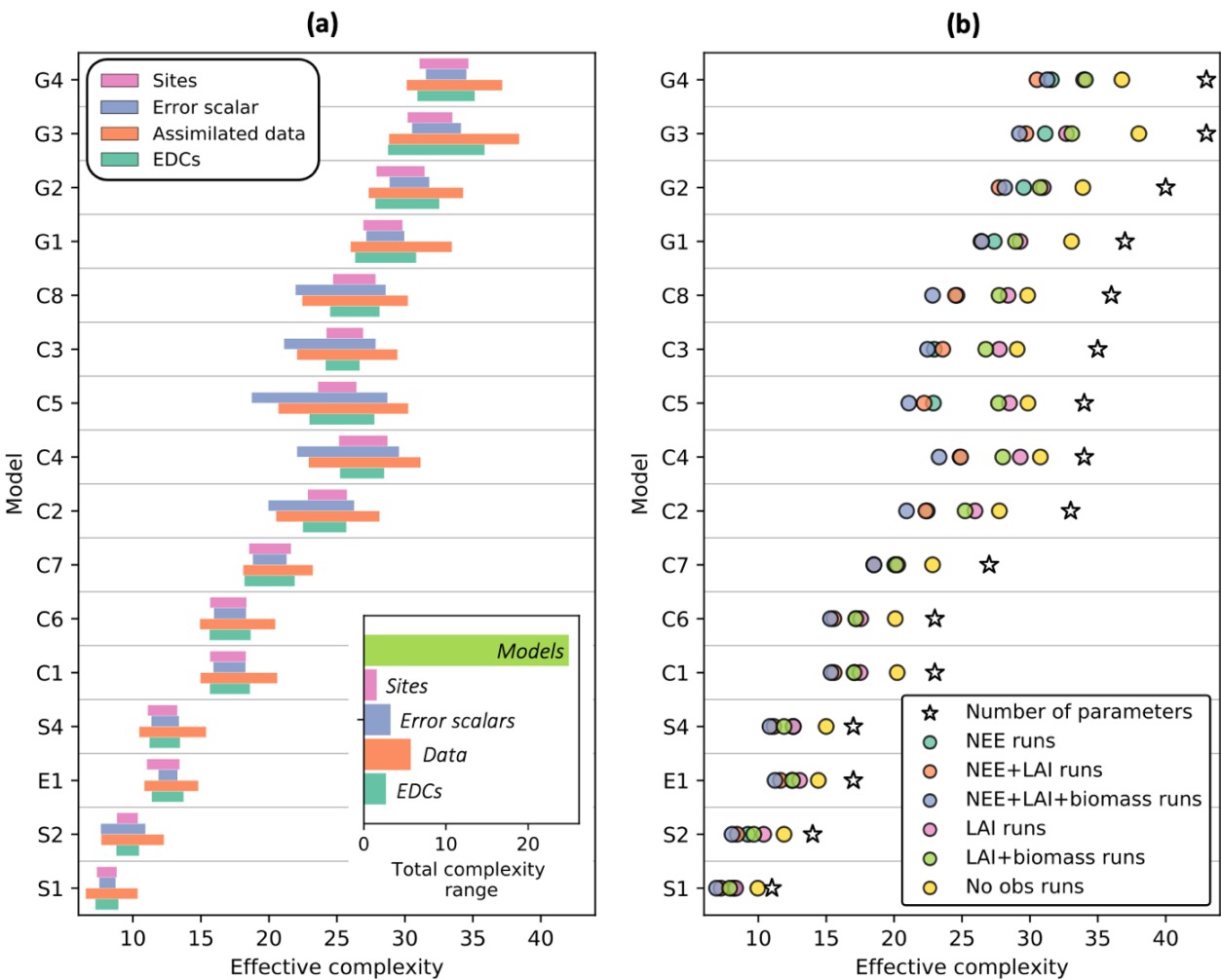

**Figure 4:** Influence of the experimentally varied factors on effective complexity. (a) Range of effective complexity attributable to sites, error scalars, assimilated data, and EDCs for each model (row). Inset: range of attributed effective complexity across all model runs. (b) Average effect of assimilated data combination on effective complexity for each model. Colored circles are means of corresponding runs. Models are ordered from fewest (S1) to greatest (G4) number of parameters. See Table 1 for definition of model IDs.

## 3.2 Relationship between effective complexity and skill

Across all runs performed in the experiment, the hypothesis that an intermediate-complexity carbon cycle model can outperform a low or high complexity model is confirmed, both when NEE is predicted *(Fig. 5a)* and when LAI is predicted *(Fig. S3a)*. Runs on both extremes of the complexity axis perform poorly, due to overfitting in the low complexity case (parameters are over-determined, leading to accurate predictions in the training period but poor ones in forecast) and





underfitting in the high complexity case (parameters are under-determined, yielding poor predictions both in training and

forecast). *Fig. 3a* and *Fig. 3c* demonstrate this contrasting behavior at the FR-LBr site.

   When runs for which no data were assimilated—that is, runs with the least informed parameters—are withheld from the

analysis, increasing complexity no longer degrades skill *(Fig. 5b)*. More specifically, the relationship between effective

complexity and skill increases monotonically when all runs have some baseline constraint on parameters. This result also holds

regardless of which variable is predicted *(Fig. S3b)* as well as when the number of runs within each complexity bin is

standardized via bootstrapping *(Fig. S4)*. This finding implies that increasing complexity by introducing suitable data-

constrained parameters can improve performance, but that doing so by adding unconstrained dimensions can degrade it. That

is, the processes and parameters introduced in the most detailed models (such as G1-G4) can lead to improvements in predictive

skill over simpler models only when they are sufficiently well-characterized (*i.e.,* adequately informed by data). Importantly,

larger observational uncertainty assumptions reduce the effectiveness of assimilated data at constraining parameters in high-

complexity models. The monotonically increasing relationship between complexity and skill is strongest when observational

error is assumed to be relatively small *(Fig. S5)*.

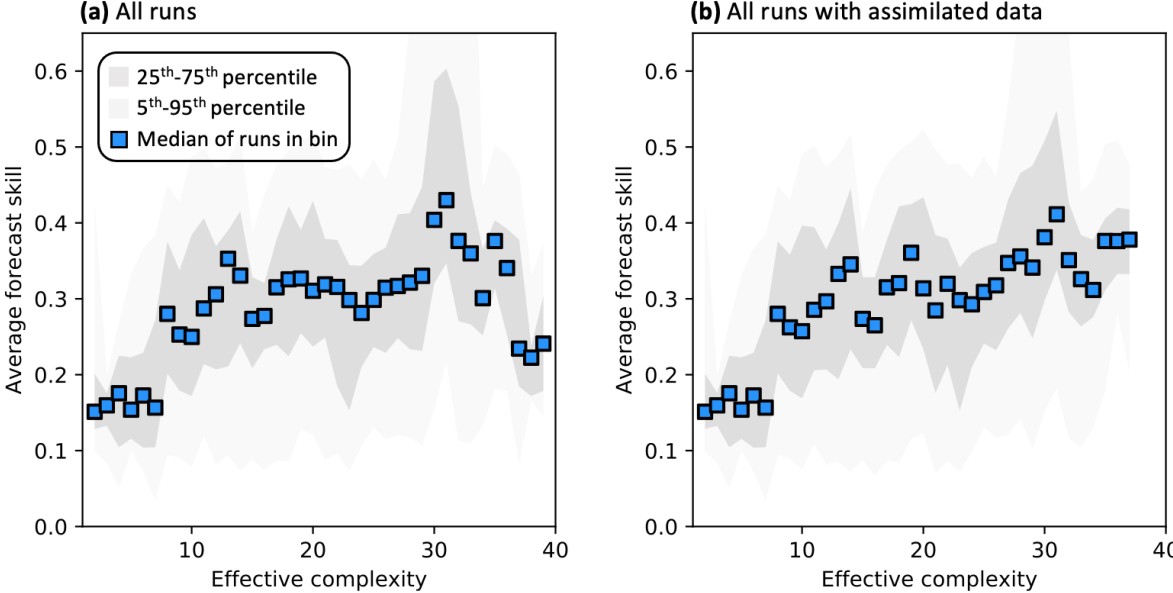

**Figure 5:** Relationship between effective complexity and NEE forecast skill for (a) all model runs in the experiment and (b)
all runs for which data was assimilated. Dark gray shading spans the 25th to 75th percentile of runs; light gray shading spans
5th to 95th percentile; blue points are medians of effective complexity bins. Average forecast skill is computed using the
histogram intersection metric.

   Assimilated data determines the shape of the overall complexity–skill relationship in the COMPLEX experiment. Not

only does the presence of any assimilated observations control the response of skill to increasing complexity, but the specific





choice of assimilated observations also matters. In particular, assimilating monthly NEE observations improves both NEE *(Fig. 6a-c)* and LAI predictions *(Fig. S6a-c)* by complex models over simple models: note the positive/increasing trends between complexity and skill in these cases. However, such improvements in predictive performance are not consistently observed across the complexity axis when other data, but not NEE, are ingested. For instance, simple models informed only by LAI perform just as well as complex models when predicting NEE. Indeed, these runs show a constant skill level across

the complexity axis *(Fig. 6d)*. The ingestion of biomass estimates in addition to the LAI data yields a small positive trend *(Fig. 6e)*, although this relationship is clearly weaker than when NEE is also assimilated *(Fig. 6c)*. When predicting LAI, though, complex models outperform simple models with only the assimilation of LAI *(Fig. S6d)*. All such combinations contrast with the case in which no data is assimilated: forecast skill for those runs declines with complexity, regardless of target variable *(Fig. 6f, Fig. S6f)*.


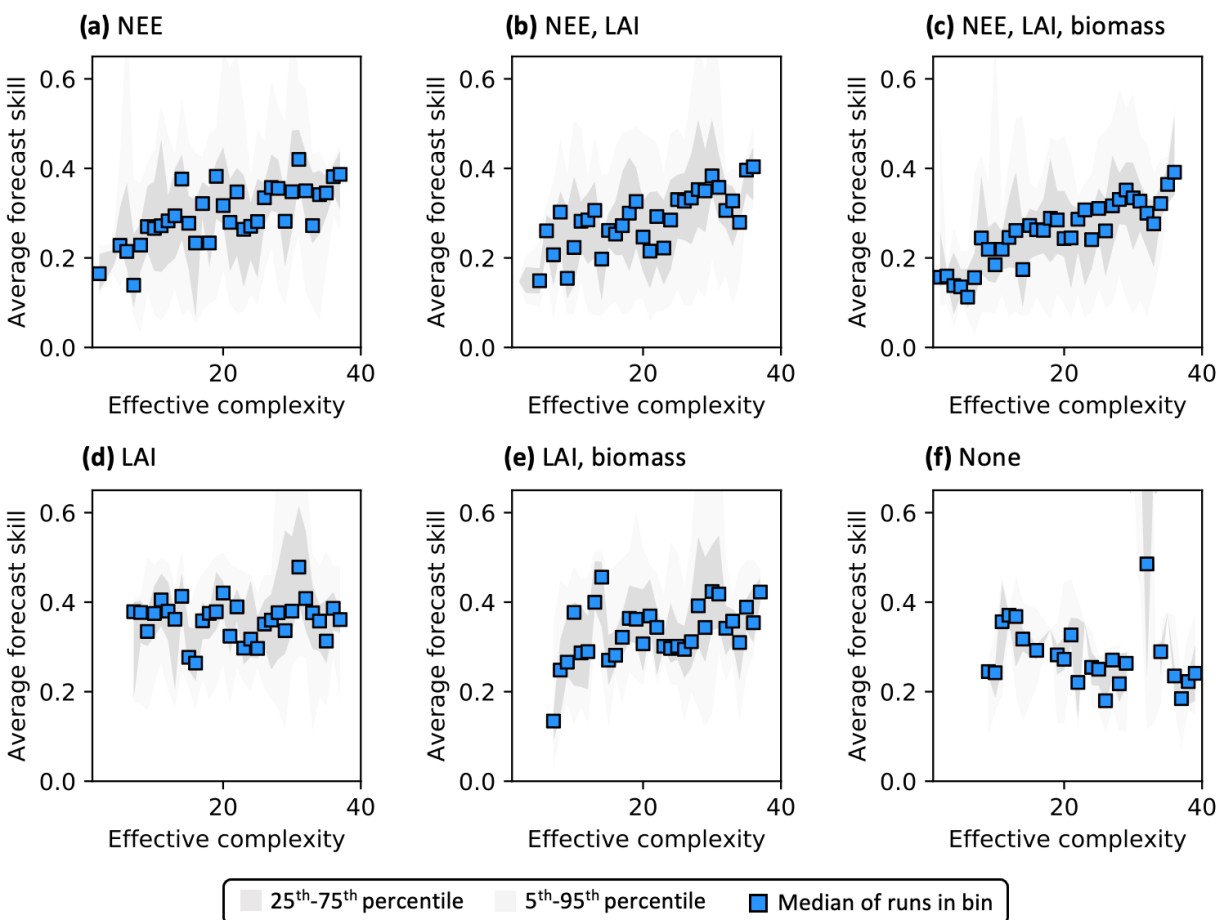

**Figure 6:** Complexity–skill relationship for NEE predictions, split by combination of assimilated data (title of each subplot). Average forecast skill is computed using the histogram intersection metric.





Recall that the magnitude of skill—the degree of overlap between model predictions and observations (see *Sect. 2.5.1*)—reflects the ability of the model to capture the data along with its uncertainty. Particularly in scenarios corresponding to low effective complexities, models tend to overfit when NEE is assimilated (as demonstrated in *Fig. 3a*). Overfitting is a key factor causing the discrepancy in performance between low-complexity runs that do *(e.g., Fig. 6a)* and do not assimilate NEE *(e.g., Fig. 6d)*.

Regardless of which data are assimilated, site-specific characteristics also introduce additional variability into the form of the relationship between effective complexity and skill *(Fig. 7)*. To better understand and isolate site-specific dynamics, here we only interpret runs for which at least one data type is assimilated. Most sites show high-complexity performance optima, consistent with *Fig. 5b*. However, several are characterized by a threshold effect for which performance increases significantly once a certain effective complexity is attained and remains stagnant thereafter *(e.g.,* a low-complexity threshold around 10 for

FI-Hyy and FR-Pue; a high-complexity threshold around 30 for US-Ha1). This "diminishing returns" effect suggests that the performance benefit of added structural detail has the potential to stabilize for all but the simplest models. The two tropical sites included in our analysis demonstrate additional unique dynamics. GF-Guy is the only site for which the performance of the most complex models appears to slightly degrade, even when all observations including NEE are assimilated, and no threshold is apparent at AU-How. Overall, the site analysis demonstrates the large variability in model performance across

space, including between sites sharing biome classifications *(e.g.,* FI-Hyy and FR-LBr) or broadly similar climate types *(e.g.,* GF-Guy and AU-How).





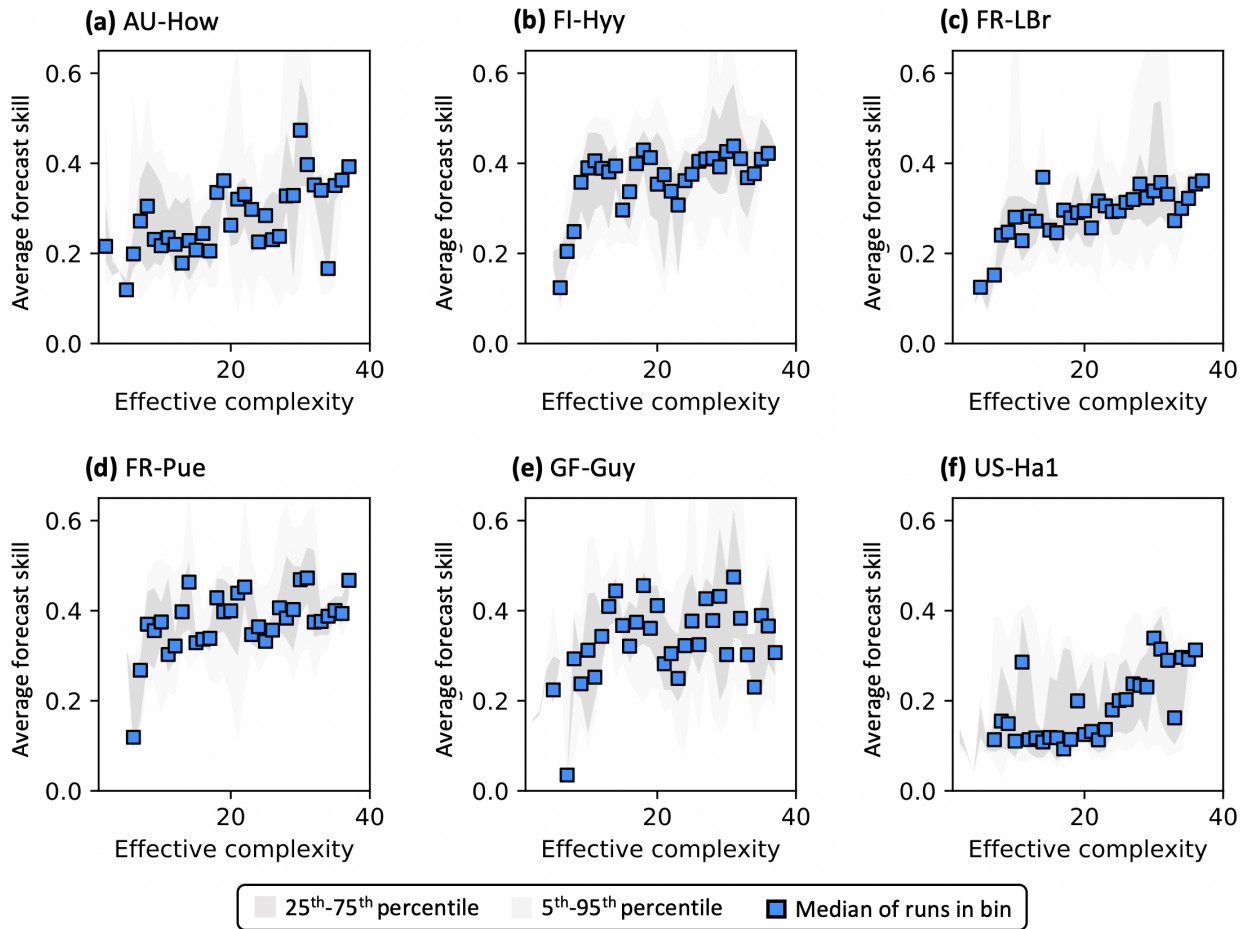

**Figure 7:** Complexity–skill relationship for NEE predictions, split by site (title of each subplot). Only runs for which data were assimilated are plotted. Average forecast skill is computed using the histogram intersection metric.


## 4 Discussion

### 4.1 Effective complexity and the inter-relationship between model structure and parameterization

We defined a concept of effective complexity that is linked to model structure and number of parameters as well as to the information content of calibration data *(Fig. 4)*. This metric can inform future studies seeking to investigate the role of model

complexity by providing a simple and comparable quantification of parameter posteriors. Conventional complexity measures (*e.g.,* counts of observable model attributes) can serve as reasonable approximations of the more nuanced definition specific to ensemble methods that we present here. Still, effective complexity is rarely identical to the number of model parameters: it is generally lower. Correlations between model parameters can and do occur whether the model is poorly- or well-constrained



*(Keenan et al., 2013)* and whether it is simple or complex, implying that all carbon cycle models have "constrainable"

dimensions. Importantly, though, none of the high-parameter models in our experiment have so much redundancy that their average effective complexity across runs is equivalent to that of any low-parameter model *(Fig. 4)*. Whether this is also true for large-scale TBMs remains an open question.

Overall, the behavior of the effective complexity metric highlights that the best-performing analyses *(i.e.,* runs with the highest forecast skill) in the COMPLEX experiment maximize model structural breadth and minimize parametric uncertainty.

Models built with high numbers of processes but without effective parameter constraints *(i.e.,* runs that maximize structural breadth but do not attempt to minimize parametric uncertainty) are not sufficient to optimize performance *(Fig. 5)*. Additionally, models of the carbon cycle can overfit if they are calibrated in too narrow a subset of conditions, and underfit if they are improperly parameterized and therefore biased, as shown in *Fig. 3.*

**4.2 Influence of data constraints and site on complexity–skill relationship**

The main factors controlling the observed complexity–skill relationship are *(a)* whether, and which, data are assimilated into the model and *(b)* the geographical location at which the analysis is undertaken. One way to interpret the role of data in the relationship is explicit: models with the ability to assimilate monthly observations of NEE, which uniquely represent the integrated behavior of terrestrial carbon cycling and its internal dynamics, are more likely to experience gains in skill with increased complexity than those that cannot. This result is consistent with the prominent role of NEE observations in reducing

model projection uncertainty identified by *Keenan et al., 2013.* The effects of LAI and biomass observations in the COMPLEX experiment are somewhat more nuanced. All models in the DALEC suite are able to extract information from the LAI data and produce reasonably skilled NEE predictions *(Fig. 6d)*, though such data do not *improve* the skill of complex models over simple ones. The ingestion of LAI data most directly constrains specific features relating to growth or carbon allocation, potentially informing the seasonality of NEE. Finally, the impacts of biomass observations on forecast skill were relatively

muted in our experiments. Given that biomass data are particularly useful for informing the carbon cycling of slow pools *(Williams et al., 2005),* the relatively short calibration (5 years) and forecast periods (≥5 years) tested here, along with the temporal sparsity of these data in the COMPLEX experiment *(i.e.,* a few measurements per site instead of continuous time series for LAI or NEE), may have obscured their utility.

Several recent TBM efforts have sought to enable the assimilation of eddy covariance or remote sensing observations

*(e.g., Bacour et al., 2015; Peylin et al., 2016; Raoult et al., 2016; Schürmann et al., 2016; MacBean et al., 2018; Norton et al., 2019)* as well as measurements of functional traits *(e.g., LeBauer et al., 2013)*. Our results underscore the value of such efforts to reduce parameter uncertainty, despite the fact that the computational costs associated with data assimilation are relatively high *(e.g., MacBean et al., 2016)*. Increased use of emulators may help reduce this computational cost *(Fer et al., 2018)*.





Given the demonstrated value of data constraints and the specification of their uncertainty *(Fig. S5)*, the need to characterize and quantify this uncertainty *(Keenan et al., 2011)* remains particularly critical for model–data fusion studies. In this analysis, NEE uncertainty was assumed to remain constant both in time *(i.e.,* for all observations regardless of season or year) and in space *(i.e.,* across sites), which likely over-generalizes the specifications of individual sensors and the possibility of systematic or increasing biases. These assumptions become even more important to account for when assimilating global

datasets, for which retrieval accuracy can vary across land cover types or with atmospheric conditions such as clouds or snow *(e.g., Fang et al., 2013)*. One benefit of the Copernicus LAI product used here is its explicit, spatially variable quantification of uncertainty, which is still relatively rare for remote sensing datasets. Though the robustness of these uncertainties has been challenged with independent observations in some locations *(e.g., Zhao et al., 2020)*, this approach represents a level of detail well-suited to the coupling of data to large-scale or global models.

The observed variability in the complexity–skill relationship across sites *(Fig. 7)* suggests that predictability itself is spatially heterogeneous. Further, it implies that the benefit to model performance accrued by the addition of a given process should not be expected to affect all locations uniformly, even when site-specific parameter uncertainty is minimized through calibration or optimization. Models not tuned locally likely smooth this spatial variability in predictability drastically *(van Bodegom et al., 2012; Berzaghi et al., 2020)*, and thus model development and calibration must include locations spanning a

wide range of vegetation, climate, soil characteristics, and disturbance histories.

**4.3 Recommendations for selecting appropriate model complexity**

Overall, our results suggest that the benefits of increased model complexity (*e.g.,* gains in skill attributable to the introduction of specific processes or to additional detail applied to existing mechanisms) are attainable only when parameters are sufficiently well characterized.  Here, this benefit is achieved when high complexity is balanced by data-assisted parameter optimization

(in particular, when NEE observations are assimilated). More broadly, the relationship between complexity and skill is dynamic and extends beyond model structural choices. As a result, it is difficult to quantify whether model parameters corresponding to any specific model implementation—including outside the DALEC suite—are adequately informed such that increased model complexity is beneficial to performance. To assist in this endeavor, we present the following recommendations for model development and evaluation:

(1)    Assimilate diverse data types to constrain model parameters at the scale of model application;

(2)    Use long time series to undertake independent forecast evaluation studies, and factor observational uncertainty into model evaluation (*e.g.* using overlap metrics);

(3)    Test whether model updates that add complexity lead to forecast improvements (not only calibration improvements), and test for possible model simplification improvements also;

(4)    Seek to calibrate or optimize model parameters even when data assimilation is not possible (*e.g.,* using optimality-based approaches; *Walker et al., 2017; Jiang et al., 2020).*





## 4.4 Transferability to large-scale models (TBMs)

This analysis tested a spectrum of structurally distinct representations of the carbon cycle based on the intermediate-complexity ecosystem model DALEC, which allowed for coupling with the CARDAMOM model–data fusion system in a computationally
tractable manner. Because our findings are not explicitly linked to the roles of specific processes or model features, however, their implications extend beyond the use of DALEC-like models to a wide variety of ecological models, including TBMs.

Traditional (PFT-based) parameter determination in TBMs is far from random. It is informed by data—for example, by hypotheses or generalizations derived from prior literature *(e.g., Oleson et al., 2010; Lawrence et al., 2011)* or by model calibration at specific locations *(e.g., Williams et al., 1997)*—and therefore endowed with ecological knowledge. Accordingly,
TBM parameters are likely more informed than the least constrained parameters retrieved in our analysis, which were freely sampled from wide uniform distributions and caused the high-complexity decline in performance *(Fig. 5)*. However, while this may be true locally, the common assumption on uniformity of parameters within PFTs casts doubt on their precision across the regional or global scales at which TBMs typically make predictions *(van Bodegom et al., 2012)*. Indeed, using a suite of global TBMs participating in the Multi-scale Synthesis and Terrestrial Model Intercomparison Project (MsTMIP; *Huntzinger*
*et al., 2013)*, Schwalm et al. (2019) showed that increases in model performance were more often linked to the omission rather than inclusion of various processes, suggesting a tradeoff between complexity and skill similar to that observed here. This conclusion calls into question the conventional paradigm that greater complexity significantly and consistently improves skill across current TBMs.

Earth observation (EO) is one key approach that can provide the high spatial and temporal resolution data on carbon
cycling needed for more localized calibrations *(Exbrayat et al., 2019)*. In COMPLEX, we used Copernicus LAI data, though there are also opportunities to ingest biomass maps from space LiDAR or radar, estimates of photosynthesis from solar induced fluorescence (SIF), and satellite-based atmospheric inversions of regional NEE, among others, in future studies. If supplied with appropriate error estimates, these datasets can over time provide powerful constraints for high resolution carbon cycle analyses with TBMs or DALEC-like models. A key research goal is to determine the appropriate model complexity for
maximizing the information content of these EO data for robust forecasts and analyses.

Alternative methodologies for deriving ecosystem parameters outside the realm of PFTs are also becoming increasingly common *(van Bodegom et al., 2012; Bloom et al., 2016; Exbrayat et al., 2018; Berzaghi et al., 2020; R. A. Fisher & Koven, 2020)* and may represent a way forward in addressing the tradeoff between structural and parametric uncertainty. Recent work has focused on upscaling in situ trait data *(e.g., from the TRY database; Kattge et al., 2020)* to yield spatially variable maps of
key ecosystem parameters, using modeled relationships with climate or canopy properties (often referred to as environmental filtering relationships, since the environment "filters" the possible distribution of parameters at a given location; *e.g., Verheijen et al., 2013; van Bodegom et al., 2014; Butler et al., 2017)*, leaf economics *(Sakschewski et al., 2015)* or optimality theory *(e.g., Smith et al., 2019)*. Other studies have investigated how TBM parameters optimized at eddy covariance sites covary with climate *(e.g., Peaucelle et al., 2019; Wu et al., 2020b)*. These efforts are not without their challenges, however. The spatial





coverage of in situ trait data as well as eddy covariance sites is sparse relative to the large diversity of ecosystem behavior *(Schimel et al., 2015)*, and such datasets also comprise a non-representative sample of species and disturbance histories *(Sandel et al., 2015)*. These biases may limit the representativeness of the modeled relationships. Taking a different approach, a small subset of models has also been developed to operate altogether independently from the paradigm of PFTs *(e.g.,* using traits-based approaches *[Pavlick et al., 2013; Scheiter et al., 2013; Fyllas et al., 2014])*. Our results imply that these and future

developments to improve the flexibility of model parameters will play critical roles in enabling the trend of increasing model complexity and may be a more fruitful avenue towards reducing the uncertainty of TBM prediction than model structural changes and additions.

## 5 Conclusions

Our approach to understanding the relationship between model complexity and model predictive performance is novel in its

focus on sampling the spectrum of possible parameter uncertainty states for a variety of model structures and calibration data. Taken together, lessons learned from the behavior of the effective complexity metric as well as the data and site effects discussed here represent a comprehensive pattern: improving the robustness of parameter calibration is a prerequisite for effectively increasing structural complexity. Specifically, we found that increasing model complexity actively degrades predictive skill in the most extreme cases of parameter uncertainty. Assimilating data—particularly monthly observations of

net ecosystem exchange—considerably improves the performance of complex models relative to simple models, though the magnitude and persistence of this improvement varies across space. Overall, the growing focus on understanding and reducing parametric uncertainties within large-scale models (such as via direct data assimilation, the development and implementation of alternatives to PFTs, parameter sensitivity analyses *[e.g., R. A. Fisher et al., 2019]*, and more) is both a necessary direction and a significant opportunity for improving the predictability of the terrestrial biosphere. Our conclusion for model

construction and usage matches those from other scientific fields, as stated by Albert Einstein: "to make the irreducible basic elements as simple and as few as possible without having to surrender the adequate representation of a single datum of experience" *(Caprice, 2013)*.



# Appendix

## Appendix A: DALEC Model Descriptions

The Data Assimilation Linked Ecosystem Carbon (DALEC) model suite includes a range of related intermediate complexity models of the terrestrial carbon cycle. Each model version is comprised of sub-models related to different simulations of photosynthesis, plant and heterotrophic respiration, canopy phenology, stomatal conductance, and the inclusion of water cycling *(Table 1)*. The sub-models are described in detail in the following sections *(Sect. A.1-A.5)*. Each section contains a table highlighting the key features of each sub-model *(Tables A1-A5)*.

### A.1.  Photosynthesis and stomatal conductance

*A.1.1.    Aggregated Canopy Model Version 1 (ACM1)*

The aggregated canopy model version 1 (ACM1) estimates canopy gross primary productivity (*i.e.,* photosynthesis) as a function of temperature, shortwave radiation, day length, atmospheric CO2 concentration, leaf area and mean foliar nitrogen content *(Williams et al., 1997; Fox et al., 2009)*. ACM1 was designed and calibrated to emulate a state-of-the-art process orientated ecosystem model SPA *(Williams et al., 1996, 2001; Smallman et al., 2013)*. As such, ACM1 contains 10 parameters which implicitly capture the more complex process representations (*e.g.,* temperature sensitivity, radiative transfer) found within SPA. An 11th parameter represents the canopy photosynthetic efficiency (the product of nitrogen use efficiency and foliar nitrogen), which is estimated by CARDAMOM as a location-specific, optimized value.

ACM1 has no explicit capacity to simulate drought or direct overheating stress on canopy processes. Canopy photosynthesis is connected to the wider carbon cycle through the leaf area, although the role of the roots in water supply is neglected as is its interplay with CO₂ supply via stomatal conductance.

*A.1.2.    Aggregated Canopy Version 1 + Cold weather GPP*

The GPP module also includes an empirical cold-weather GPP limitation sensitivity function. The cold temperature limitation factor (denoted as *g*) is used as a multiplier on the DALEC GPP function output, to act as a thermostat that regulates evergreen needleleaf carbon uptake. The cold-weather factor *g* is calculated using added model parameters ($T_{minmin}$ and $T_{minmax}$) and temperature observations ($T_{min}$), such that $g = 0$ if $T_{min} < T_{minmin}$, $g = 1$ if $T_{min} > T_{minmax}$, and $g = (T_{min} - T_{minmin})/(T_{minmax} - T_{minmin})$ otherwise.

*A.1.3.    Aggregated Canopy Version 2 (ACM2)*

The aggregated canopy model for gross primary productivity and evapotranspiration is the successor version to ACM1, hereafter known as ACM2 *(Smallman & Williams, 2019)*. ACM2 builds on the ACM1 outline creating a model of ecosystem water cycling to facilitate the implementation of a mechanistic stomatal conductance model





linking the canopy to soil water via fine roots and optimizes the stomatal intrinsic water use efficiency (for details see *Williams et al. [1996]* and *Bonan et al. [2014])*. ACM2 simulates shortwave and longwave isothermal radiation balances, canopy interception of rainfall and soil infiltration. ACM2 is therefore capable of simulating canopy

transpiration, soil evaporation, evaporation of canopy intercepted rainfall, soil water runoff and drainage.

*A.1.4.    Analytical Ball-Berry*

For the analytical Ball-Berry GPP module of CARDAMOM, leaf-level GPP and stomatal conductance are calculated using the coupled leaf photosynthesis-stomatal conductance developed by Ball-Berry (*Ball et al., 1987*) and an

analytical solution to the system of equations developed by Baldocchi (*Baldocchi, 1994*). This new module serves to both calculate GPP and evapotranspiration coupled through the stomatal behavior. This formulation added the maximum rate of carboxylation ($V_{cmax}$), the maximum rate of electron transport ($J_{max}$), stomatal slope and intercept, and boundary layer conductance to the set of parameters that were optimized through data assimilation, while removing the explicit water use efficiency (where there is a water cycle in CARDAMOM) and canopy efficiency

parameters. We scaled the leaf level results of GPP and stomatal conductance to the canopy as a 'big leaf' with an exponential decay function of LAI (*Sellers et al., 1992*).

**Table A1:** Summary of the key features for each photosynthesis sub-model

| Sub-Model | Key Feature(s) |
|---|---|
| ACM1 | 1. Estimates GPP sensitive to temperature, $CO_2$, SW radiation, leaf area<br>2. Stomatal conductance uses empirical approach |
| ACM1 + Cold weather GPP | Same as ACM1, includes an empirical cold-weather GPP suppression scheme |
| ACM2 | 1. Estimates GPP and ET sensitive to temperature, $CO_2$, SW radiation, leaf area, water supply via fine roots<br>2. Stomatal conductance uses optimality approach<br>3. Simulates full ecosystem water balance |
| Analytical Ball-Berry | 1. Sensitive to temperature, $CO_2$, SW radiation, leaf area<br>2. Stomatal conductance uses empirical approach<br>3. Simulates full ecosystem water balance<br>4. Time-varying water use efficiency |


**A.2.  Autotrophic respiration ($R_a$)**

Autotrophic (plant) respiration (Ra) is a key ecosystem carbon flux returning approximately half of GPP back to the atmosphere *(Waring et al., 1998)*. While this overall proportionality remains true, subsequent studies have identified variation in the



$R_a$:GPP fraction linked, among others, to climate, nutrient status and plant age *(e.g., Collalti & Prentice, 2019)*. Furthermore,

there are multiple competing hypotheses for how to explain the broad proportionality and site-specific variations *(e.g., Collalti & Prentice, 2019; Collalti et al., 2020)*, requiring an investigation of multiple approaches.

### A.2.1.   *Fixed $R_a$:GPP fraction*

Autotrophic respiration ($R_a$) is assumed to be a fixed (time-invariant) fraction of GPP ($R_a$:GPP) such that

$$R_a = GPP \times R_a : GPP \qquad\qquad (A1)$$

It varies in space as a retrieved location specific parameter. A prior value ($0.46 \pm 0.12$) for the $R_a$:GPP fraction is drawn from *Waring et al. (1998)* and *Collalti & Prentice (2019)*.

### A.2.2.   *Fixed $R_m$:GPP fraction $R_g$:NPP*

$R_a$ can be divided between respiration associated with tissue growth ($R_g$) and maintenance ($R_m$). $R_g$ has a robust mechanistic understanding, allowing it to be estimated as a fixed fraction of carbon allocated to plant tissues ($C_{alloc}$; gC/m$^2$/d) independently of ecosystem type and climatic conditions (0.22; *Waring & Schlesinger, 1985)*:

$$R_g = C_{alloc} \times 0.22 \qquad\qquad (A2)$$

We continue to retrieve a location specific fixed fraction of GPP respired as $R_m$ ($R_m$:GPP):

$$R_m = GPP \times R_m : GPP \qquad\qquad (A3)$$

This formulation allows for variation between the proportion of $R_a$ attributed to either $R_g$ or $R_m$, as they have independent drivers. Note that this model structure implicitly assumes that maintenance respiration is fully coupled to GPP and growth activity, neglecting any distinct temperature sensitivity of respiration versus photosynthesis.

### A.2.3.   *Canopy Cost Respiration Model*

The sensitivity of $R_m$ to tissue temperature and nitrogen content is well established *(e.g., Ryan, 1991; Reich et al., 2008; Atkin et al., 2017)*, however the exact formulation of the relationship remains poorly understood *(Thomas et al., 2019)*. We implemented the canopy maintenance respiration model proposed by *Reich et al. (2008),* which has been extensively evaluated in comparison with alternate approaches *(Thomas et al., 2019)*. Wood and fine root

maintenance respiration continue to be represented using a fixed fraction as described in *Sect. A.2.2*. Estimation of growth respiration continues to be a fixed fraction of NPP.

Following *Reich et al. (2008)*, the estimation of canopy maintenance respiration occurs in two stages: *(i)* estimation of the canopy maintenance respiration per gram leaf carbon at 20ºC ($R_{m\text{-}leaf}^{20}$; gC/m$^2$leaf/d); and *(ii)* daily temperature adjustment. $R_{m\text{-}leaf}^{20}$ is estimated as a function of the leaf nitrogen concentration ([$N_{leaf}$]; mmolN/gleaf)

and two retrieved parameters. Parameter $\alpha$ represents the reference maintenance respiration at 20°C and [$N_{leaf}$] = 1, while $\beta$ is the exponential [$N_{leaf}$] sensitivity parameter. Both $\alpha$ and $\beta$ are retrieved by CARDAMOM as DALEC model parameters. The *Reich et al. (2008)* model estimates maintenance respiration in units of nmolC/gleaf/s, which





is adjusted to gC/gCleaf/d by the remaining terms: $1 \times 10^{-9}$ scales from nmolC to molC; 12 is the atomic mass of carbon adjusting molC to gC; the factor 2 adjusts gC/gleaf/s to gC/gCleaf assuming 50 % of leaf biomass is carbon; and 560 86400 is the number of seconds in a day giving gC/gCleaf/d:

$$R_{m-leaf}^{20} = 10^{\alpha} \times [N_{leaf}]^{\beta} \times (1 \times 10^{-9}) \times 12 \times 2 \times 86400 \qquad (A4)$$

[$N_{leaf}$] is determined from existing DALEC parameters representing the mean foliar nitrogen content (avN; $gN/m^2$) and leaf mass per unit area (LMA; $g/m^2$):

$$[N_{leaf}] = \left(\frac{avN/LMA}{14}\right) \times 1000 \qquad (A5)$$

The factor of 14 is the atomic weight of nitrogen and 1000 scales from to mmolN.

Temperature strongly impacts metabolic activity and thus maintenance respiration. The canopy maintenance respiration ($R_{m-leaf}$) at the current temperature (T) is estimated following a Q10 function (=2; widely used) and scaled by the size of the canopy carbon pool ($C_{fol}$; $gC/m^2$):

$$R_{m-leaf} = R_{m-leaf}^{20} \times 2^{0.1(T-20)} \times C_{fol} \qquad (A6)$$

The instantaneous temperature response is well captured by existing models. However, the impact of long-term temperature changes and associated acclimation of both photosynthetic and respiratory pathways is not accounted for. Therefore, simulations over longer time scales may overestimate negative feedbacks of increased canopy maintenance respiration due to warming *(Atkin et al., 2015; Wang et al., 2020)*.

**Table A2:** Summary of key features for each respiration sub-model.

| Sub-Model | Key Feature(s) |
|---|---|
| Fixed $R_a$:GPP | 1. Simple approach supported by literature on annual timescales |
| Fixed $R_m$:GPP + $R_g$:NPP | 1. Simple approach with well supported literature values for growth respiration ($R_g$) <br> 2. Allows quantification of relative importance of growth and maintenance respiration ($R_m$) |
| Canopy Cost Respiration Model | 1. Links canopy respiration to traits and temperature <br> 2. Facilitates implementation of economic models of canopy phenology |

**A.3. Decomposition and heterotrophic respiration**

Heterotrophic respiration results from decomposition and mineralization processes carbon pools containing dead organic 580 matter. Depending on the model structure, these can include a fine litter pool ($R_{h-lit}$ composed of foliar and fine root inputs), a





wood litter ($R_{h\text{-woodlit}}$ both fine and coarse woody debris) and soil organic matter ($R_{h\text{-som}}$). In all cases, decomposition and mineralization follow a first order kinetic approach with environmental modifiers. When litter and wood litter pools turn over, a fraction of their carbon is released as heterotrophically respired C while the remainder passes to the soil organic matter pool ($D_{lit}$, $D_{litwood}$; $gC/m^2/day$). All decomposition of soil organic matter is heterotrophically respired. All models assume
heterotrophic C respiration is respired as $CO_2$.

*A.3.1.   Temperature sensitivity*

All dead organic matter pools follow a common basic form of a pool specific turnover parameter ($\Theta_{pool}$; fraction per day at 0°C) combined with an exponential response linked to temperature (Tmax; C) and a sensitivity parameter ($\gamma$):

$$R_{h-pool} = C_{pool} \times \Theta_{pool} \times e^{\gamma T_{max}} \qquad (A7)$$

*A.3.2.   Temperature and soil moisture sensitivity*

Heterotrophic respiration regulated by both temperature (as in *Sect. A.3.1*) and a linear function of the ratio of current precipitation to the site mean (as proxy for near-surface soil moisture). The functional form allows for varying linear
sensitivity, such that:

$$R_{h-pool} = C_{pool} \times \Theta_{pool} \times f(T) \times \left( \left( (P/\underline{P}) - 1 \right) * s_p + 1 \right) \qquad (A8)$$

where P is the monthly precipitation, $\underline{P}$ is the average precipitation, and $s_p$ is the precipitation sensitivity parameter. Note that sensitivity is positive-definite (*i.e.,* no heterotrophic limitations induced for high moisture events). See *Quetin et al. (2020)* and *Bloom et al. (2020)* for further detail.


**Table A3:** Summary of key features for each decomposition sub-model.

| Sub-Model | Key Feature(s) |
|---|---|
| Temperature sensitivity | 1.  Robust estimation of 1st order exponential temperature sensitivity |
| Temperature and soil moisture sensitivity | 1.  Robust estimation of 1st order exponential temperature sensitivity<br>2.  Varying linear sensitivity to moisture content |

## A.4.  Canopy phenology

*A.4.1.   Combined Deciduous-Evergreen Analytical (CDEA) model*

The CDEA phenology model is based primarily on a day of year approach to simulate the turnover of a labile pool to support canopy growth and subsequent canopy turnover *(Bloom & Williams, 2015)*. Each timestep, a fixed fraction of GPP is allocated to the canopy and a labile pool which supplies the canopy with new growth based on the CDEA model. The CDEA model uses parameterized values for the peak day of year for labile turnover (*i.e.,* supplying leaf



growth) and leaf turnover plus two further parameters which define the standard deviation of a Gaussian distribution
specifying the period of time over which canopy phenology occurs. The fraction of the canopy which is turned over
each year is defined by a leaf lifespan parameter, while the labile pool is assumed to fully turnover each year.

The CDEA model provides an easy to calibrate diagnostic model of mean canopy phenology. However, it does
not vary phenology in response to changing environmental conditions limiting simulation of inter-annual variability.
As a result, the CDEA model has a limited capacity to inform on the meteorological drivers of canopy phenology.

*A.4.2.* *CDEA+*

Phenology same as *Sect. A.4.1*; labile C release to foliar C is optimizable (annually ~15-100% of labile C allocated
to foliar C).

*A.4.3.* *Growing Season Index (GSI) + GPP return*

Canopy phenology is sensitive to environmental conditions *(e.g., Jolly et al., 2005; Forkel et al., 2015)* and plant
carbon economic constraints *(e.g., Flack-Prain et al., 2020)* driving interannual variation of leaf area dynamics. The
growing season index (GSI) is a piecewise model linking canopy phenology to linear functions of day length,
temperature and vapor pressure deficit scaled 0-1 (GSI; *Jolly et al., 2005*). The GSI model was implemented in
*Smallman et al. (2017)* and augmented to include a requirement for new leaf area to lead to an increase in GPP greater
than a critical threshold retrieved as part of CARDAMOM.

However, we note that recent plant economic theory indicates that canopies are optimizing net canopy carbon
export (NCCE; *e.g., Thomas et al., 2019; Flack-Prain et al., 2020*)—that is, photosynthesis less respiratory and
construction costs, rather than photosynthesis alone. To investigate this level of process complexity, in *Sect. A.4.4* we
include a canopy maintenance respiration model to assess the NCCE.

*A.4.4.* *Growing Season Index (GSI) + Net Canopy Carbon Export (NCCE)*

Optimality theory is increasingly being used to explain canopy phenology based on maximizing some metric of the
carbon economy. One approach which is gaining support is optimizing net canopy carbon export (NCCE): that is,
ensuring photosynthetic gains are greater than costs associated with leaf growth and maintenance respiration *(e.g.,
Thomas & Williams, 2014; Flack-Prain et al., 2020).* While further research is needed to refine these theoretical
models, we implement a model consistent with existing literature.

The GSI model proposes an amount of new leaf area. Whether this grows or not is determined by quantifying
whether the increase of GPP averaged over the expected life span of the leaf is greater than the increased maintenance
respiration costs and the carbon required to construct the new leaf and the associated growth respiration.





**Table A4**: Summary of key features for each phenology sub-model.

| Scheme | Key Feature(s) |
|--------|----------------|
| CDEA | 1. Simple to calibrate, provides robust diagnostic of canopy phenological timing |
| CDEA+ | 1. Same as CDEA, with variable labile release fraction |
| GSI | 1. Links canopy phenology to environmental factors supporting prognostic simulations |
| NCCE | 1. Links canopy phenology to environmental factors supporting prognostic simulations<br>2. Introduces economic return on canopy investment. |

## A.5. Water cycling

### A.5.1. Empirical Bucket

The bucket approach extends the DALEC baseline structure to include a plant-available water pool, where the hydrological balance is defined as the sum of precipitation inputs (P) and evapotranspiration (ET) and runoff (R) outputs. The total plant-available water W at time $t$+1 is determined in the following way:

$$W(t + 1) = W(t) + \big(P(t) - ET(t) - R(t)\big)\Delta t \tag{A9}$$

where $\Delta t$ is the time period. Runoff is calculated as

$$R(t) = \alpha W(t)^2 \tag{A10}$$

where $\alpha$ is a second-order decay constant. Evapotranspiration is derived as

$$ET(t) = GPP(t)\frac{VPD(t)}{\nu_e} \tag{A11}$$

where $\nu_e$ is the inherent use efficiency. The plant-available water limits GPP such that

$$GPP(t) = GPP_{max}(t) \cdot max\left(1, \frac{W_t}{\omega}\right) \tag{A12}$$

where $\omega$ is the plant-available water stress threshold. Note that the parameters $\alpha$, $\nu_e$, $\omega$, and $W_0$ are optimized in CARDAMOM. For further details, see *Quetin et al. (2020)* and *Bloom et al. (2020).*

### A.5.2. ACM2: Multi-layer root model

The ACM2 model includes a multi-layer representation of the soil and root access *(Smallman & Williams, 2019)*. There are 5 soil layers, three of which are accessible to roots to supply the canopy with water. The top two layers have a fixed thickness of 10 and 20 cm respectively with a third layer which is expandable based on root penetration. Soil layer specific field capacity, porosity and hydraulic conductances are calculated using soil texture. Using these



data, infiltration of precipitation, drainage between soil layers, soil hydraulic resistance to root uptake of water and
soil surface evaporation are estimated. Soil surface evaporation occurs from the top soil layer only. For a complete
description, see *Smallman & Williams (2019)*.

**Table A5:** Summary of key features for each water cycle sub-model.

| Scheme | Key Feature(s) |
|---|---|
| Empirical Bucket | 1. First-order plant-soil carbon-water feedback |
| ACM2: Multi-layer root model | 1. Allows semi-mechanistic representation of hydraulic processes<br>2. Explicit representation of transpiration, wet canopy evaporation, soil evaporation, drainage and runoff |


**Appendix B: Carbon Cycle Structure for DALEC Variants**

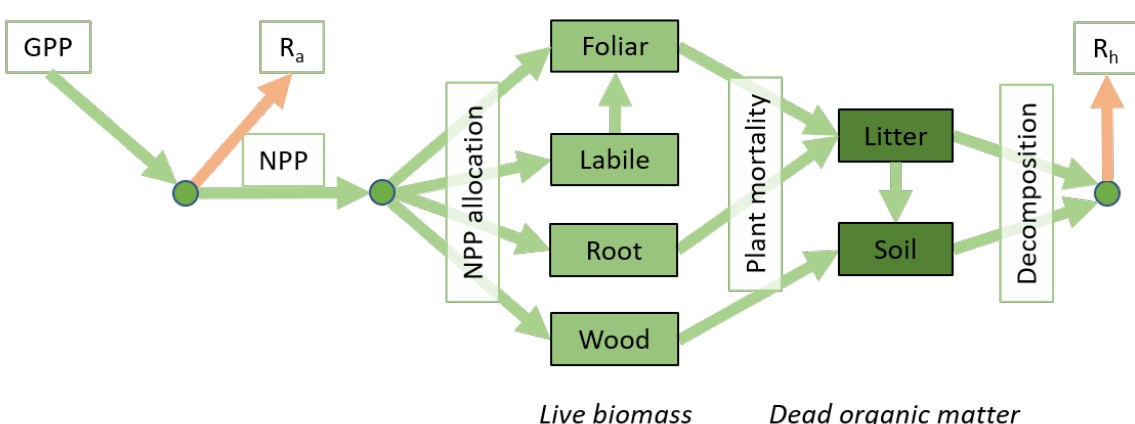

**Fig. B1:** Carbon cycle structure for models C1-C8.



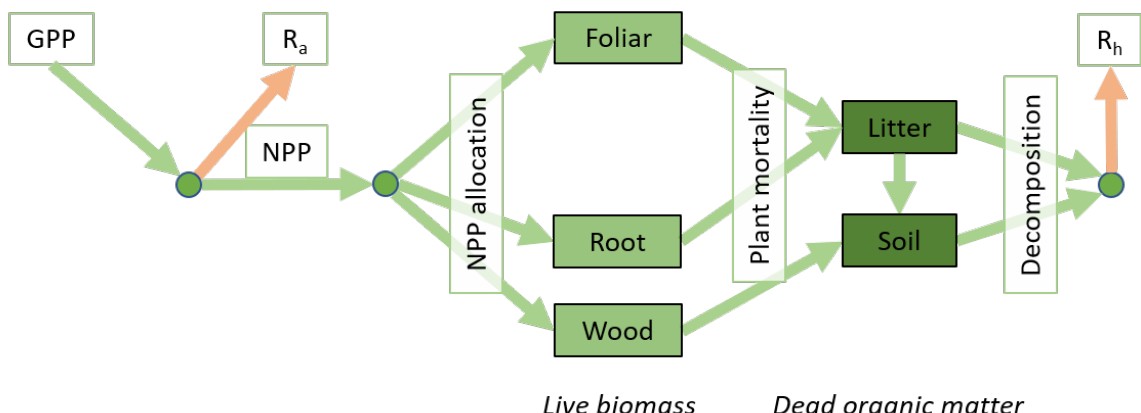

**Fig. B2:** Carbon cycle structure for model E1.

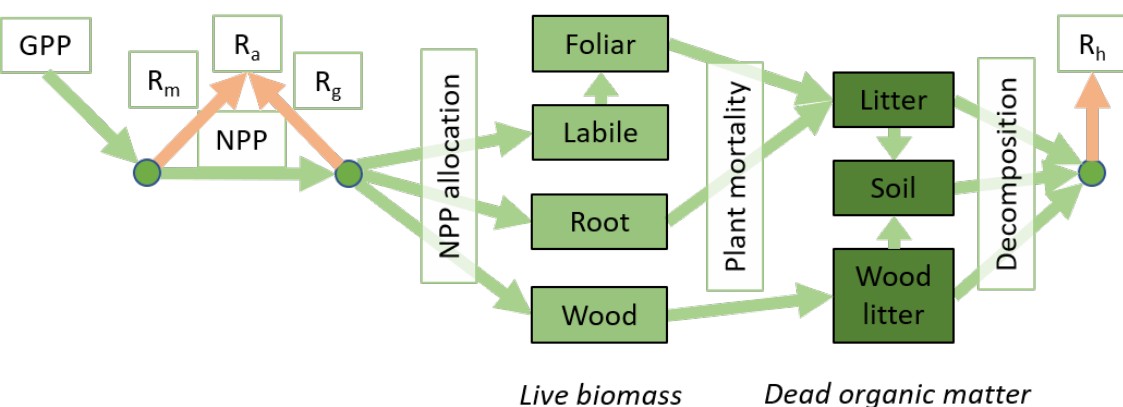

**Fig. B3:** Carbon cycle structure for models G1-G4.




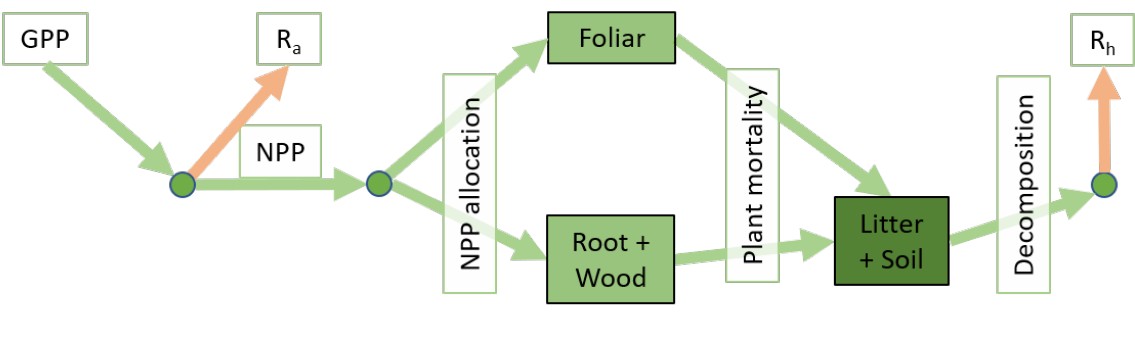

**Fig. B4:** Carbon cycle structure for model S1.

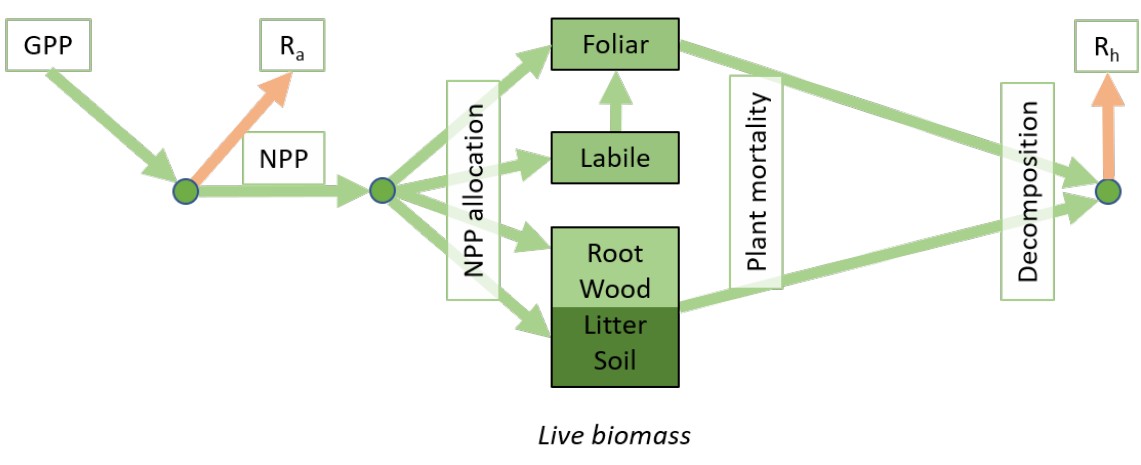

**Fig. B5:** Carbon cycle structure for model S2.



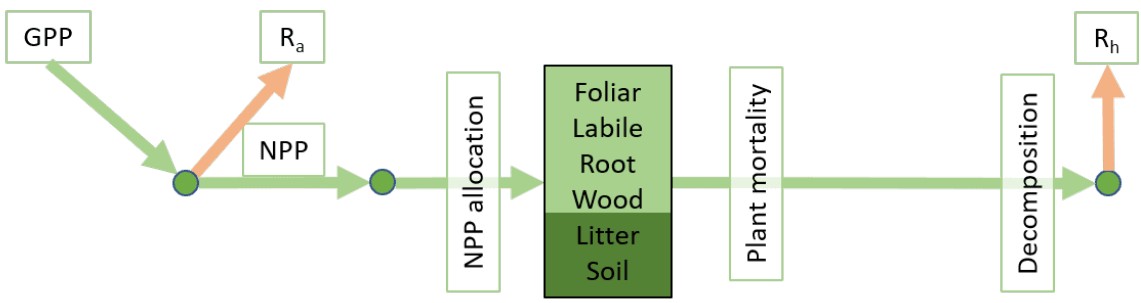

**Fig. B6:** Carbon cycle structure for model S3.

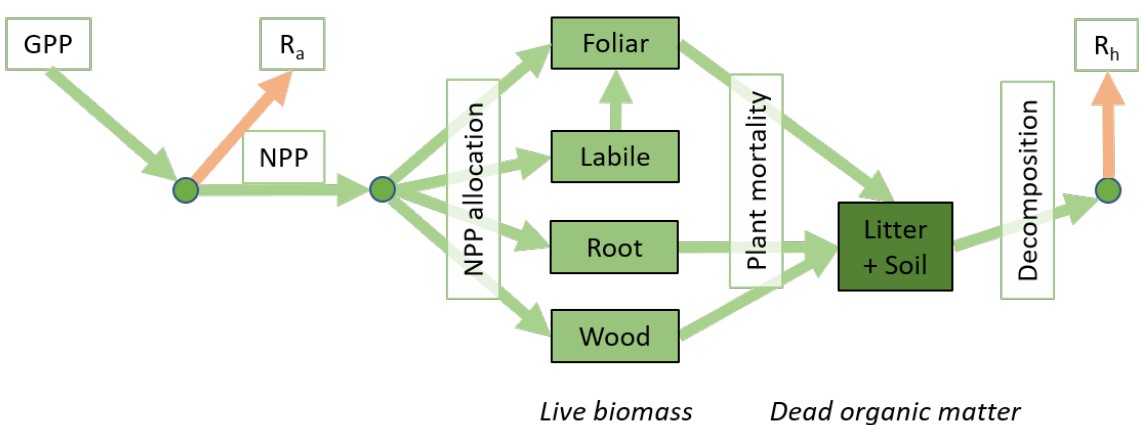


**Fig. B7:** Carbon cycle structure for model S4.

**Appendix C: Data Requirements and Site Selection**

The COMPLEX experiment uses information from 6 sites across the globe *(Fig. C1)*. The selection aimed to maximize their

biogeographical spread and diversity of natural ecosystems while fulfilling specific data requirements. A key DALEC model criterion requires that the sites must not be dominated by C4 photosynthetic pathway, be arable agriculture or intensively grazed grassland. The COMPLEX experiment makes use of a range of time series observations, including LAI, NEE and wood stock inventory. Furthermore, the experiment uses temporally distinct calibration and prediction periods requiring



observational constraints to span both periods. Collectively both scientific and data availability created a series of site selection

criteria which are described below.

Time series information on leaf area are drawn from the Earth Observation (EO) derived Copernicus 1 km product which provides estimates of LAI magnitude at fine temporal resolution and concurrent location specific estimate on uncertainty. Using this EO product and the above-mentioned calibration / prediction period constraints requires sites data collection periods to be post 1998.

Simulation of NEE is a key focus of the COMPLEX experiment, making the availability of long-term, temporally consistent, high quality NEE estimated derived from eddy covariance essential *(e.g.*, FLUXNET2015; *Pastorello et al., 2020)*. The FLUXNET2015 database provides consistent information on data quality *(e.g.,* observation uncertainty and proportion of model-data gap-filling) that underpin the site selection process. Here, to avoid comparing DALEC-simulated NEE with largely statistically gap-filled observations, only sites with < 20% gap-filled data are used.

*Hill et al. (2012*) demonstrated that assimilation of NEE observations provides substantial new information up to at least 5 years in duration. To create a balanced experimental design, COMPLEX sites are required to have a minimum of 10 years of observations *(i.e.,* 5 years calibration and remainder evaluation). Building on existing analyses with DALEC *(e.g., Smallman et al., 2017)*, COMPLEX quantifies the role of woody biomass information on constraining the DALEC models' predictive capacity of NEE. Therefore, multiple wood stock estimates are required spanning both the calibration and prediction

periods. As determining the amount and accessing of inventory data often requires direct contact with site managers, this stage occurs later in the selection process.

Collectively, the above mentioned and model process representations formed the basis of a site selection procedure to filter the FLUXNET2015 database. This process ultimately led to the selection of 6 sites *(Table 2)*.

(a)   Sites must represent a natural ecosystem *(i.e.,* remove arable agriculture and intensively grazed sites) dominated by
C3 photosynthesis species.

(b)   Sites have observations spanning > 10 years after 1998.

(c)   Sites have < 20% gap-filled observations: threshold varied to ensure that at least one site representative is available for boreal, temperate and tropical ecosystems spanning, where appropriate, canopy phenological types *(i.e.,* needle versus broadleaf, evergreen versus deciduous).

(d)   Contact site managers to determine availability of wood stock observations.





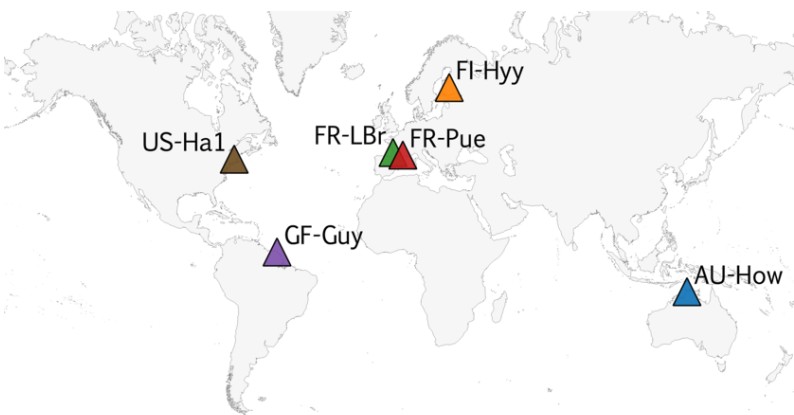

**Fig. C1:** Map of FLUXNET sites used in the experiment.

**Acknowledgments**

*Contributions.* A.A.B., M.W., T.L.S., A.G.K., G.R.Q., S.F.-P., V.M., and C.A.F. planned the analysis. C.A.F., T.L.S., P.A.L., G.R.Q., S.F.-P., V.M., N.C.P., S.G.S., Y.Y., A.A.B., M.W., and A.G.K. contributed to model development. T.L.S. and M.W. developed site selection criteria, contacted site PI's, and gathered input data. T.L.S. and P.A.L. executed model runs. C.A.F. performed analysis on model outputs. C.A.F. wrote the manuscript with contributions from T.L.S., P.A.L., G.R.Q.,
A.A.B., M.W., and A.G.K. All authors reviewed drafts of the manuscript.

*Data Availability.* Data generated in the COMPLEX experiment (performance and complexity metrics corresponding to each model run) are publicly available at doi.org/10.6084/m9.figshare.13409096. We thank FLUXNET site PIs Jean-Marc Ourcival and Serge Rambal (FR-Pue), Lindsay Hutley and Jason Beringer (AU-How), Bill Munger and Steve Wofsy (US-Ha1), Denis Loustau (FR-LBr), Timo Vesala (FI-Hyy) for providing much of the data used in our analysis. We thank Yuan
Zhao, Rong Ge and Penghui Zhu for their assistance in preparing the data.

*Funding.* M.W. acknowledges funding from NERC (NE/P018920/1), UK Space Agency, Newton Fund CSSP Brazil, and the Royal Society. C.A.F., G.R.Q., and A.G.K. were supported by NSF DEB-1942133. Operation of the US-Ha1 site is funded by the U.S. Department of Energy's Office of Science (DE-AC02-05CH11231), and National Science Foundation LTER funding (DEB-1832210). The Howard Springs site is funded by Australian Research Council FT1110602, DP160101497
and Australian Terrestrial Ecosystem Research Network – Ecosystems Process platform. This work has made use of the resources provided by the Edinburgh Compute and Data Facility (ECDF) (http://www.ecdf.ed.ac.uk/).

*Conflicts of Interest.* The authors declare they have no conflicts of interest.





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
