# Peer review of "Optimal model complexity for terrestrial carbon cycle prediction"

_Biogeosciences, 2020_

## Referee Comment (RC1) · Anonymous Referee #1 · 28 Jan 2021

The paper by Famiglietti et al uses a suite of data assimilation (parameter optimization) experiments that encompasses models of varying degrees of complexity together with different datasts included in the assimilation to test to what degree model complexity impacts model forecast skill. This is motivated by the general, but not widely tested, assumption that increasing model complexity (and in doing so, the number of parameters) may increase model realism but decrease model predictive skill.

Crucially in this DA context, they use a complexity metric that accounts for both the model structural complexity and the information content of the data that are used to optimize the model parameters.

They show that when unconstrained by data, models of intermediate complexity have the highest skill, thus demonstrating a trade-off between complexity and skill. However,

they nicely demonstrate that when constrained by data, models of higher complexity also achieve high forecast skill; thus, confronting models against data (i.e. calibrating parameters, or constraining parameters by some other method) is a prerequisite for increasing the complexity of model structure.

This is important work, particularly given the global terrestrial biosphere model community is still striving to increase the representation of different processes that are deemed necessary to realistically simulate the impacts of climate and environmental change. However, the same community is not investing heavily in implementing DA to constrain uncertainty in their models. This study proves that the two should come hand in had.

I thought the study was well designed and executed, and the results clearly described and nicely discussed. I only have a few thoughts and suggestions:

In Section 2.5.1 I would explicitly state why you are using this complexity metric (i.e. that it links both the model complexity and number of parameters but also the information content of data), instead of the other model complexity metrics that are available. This is stated multiple times elsewhere, but I think it would be useful here as well.

I'm not 100% sure what point I'm trying to make with this comment so bear with me, but I found myself wanting to dig into more of the nuances of Figure 4, especially in relation the different processes that are included in the model and the level of detail for each process. You do explore more about the of the differences related to the type of data that are included when you talk about Fig 6, and about the differences across sites with Fig 7. So I found myself wondering about the impact process representation (e.g. ACM v1 vs v2). But I appreciate that's beyond the scope of this study. I guess it just may be worth pointing out (the relatively obvious point) that this framework could also be used to determine which exact model representation is most useful for representing a given dataset – e.g. whether to include a water cycle, or not etc. Minor comments

It would be good to add the prior to Fig. 3, just to see how well the DA system is doing.

Unless I am misunderstanding the histogram intersection metric you have described in
2.5.1 would not range between 0 and 1. Perhaps you mean the normalized intersection
metric?

―――――――――――――――――――――――――

---

## Referee Comment (RC2) · Enqing Hou (Referee) · 15 Feb 2021

Famiglietti and colleagues explored relationship between model complexity and forecast skill either with or without assimilated data using a data assimilation system. The authors found that without assimilated data, a complex model has a poorer forecast skill than a simple model; with assimilated data, the opposite is true. The findings make sense and highlight the importance of using data to inform model before forecasting. The manuscript is very interesting and well written. I have only a few minor concerns about the manuscript below. L230-232: Will there be any difference in key results and conclusion obtained between using the histogram interaction and using the more familiar metrics? L233-239: n value (number of bins) used is? L405: "assimilate diverse data types" operates blindly. Some datasets are more useful to constrain

a specific variable than other datasets. We can do better than just "diverse". Table 1. Explain the meanings of the IDs (e.g., C groups, S groups). Why the sub-models ordered in the current way in the Table? Fig. 5a: "(a) All runs", do you mean all runs without assimilating data? You may make it clearer. Fig. 6: arrangement of the panels are not in a good logic to me. Probably as (a) None ... (f) NEE, LAI, biomass. Description of skill metric and complexity metric as well as the model structure are clear, but deposit the code to produce this manuscript could be more helpful to others to use the approaches here.

---

## Author Comment (AC1) · 24 Feb 2021

**Author response to interactive comments on "Optimal model complexity for terrestrial carbon cycle prediction"**

**Reviewer:** The paper by Famiglietti et al uses a suite of data assimilation (parameter optimization) experiments that encompasses models of varying degrees of complexity together with different datasets included in the assimilation to test to what degree model complexity impacts model forecast skill. This is motivated by the general, but not widely tested, assumption that increasing model complexity (and in doing so, the number of parameters) may increase model realism but decrease model predictive skill.

Crucially in this DA context, they use a complexity metric that accounts for both the model structural complexity and the information content of the data that are used to optimize the model parameters.

They show that when unconstrained by data, models of intermediate complexity have the highest skill, thus demonstrating a trade-off between complexity and skill. However, they nicely demonstrate that when constrained by data, models of higher complexity also achieve high forecast skill; thus, confronting models against data (i.e. calibrating parameters, or constraining parameters by some other method) is a prerequisite for increasing the complexity of model structure.

This is important work, particularly given the global terrestrial biosphere model community is still striving to increase the representation of different processes that are deemed necessary to realistically simulate the impacts of climate and environmental change. However, the same community is not investing heavily in implementing DA to constrain uncertainty in their models. This study proves that the two should come hand in hand.

I thought the study was well designed and executed, and the results clearly described and nicely discussed. I only have a few thoughts and suggestions.

**We thank the reviewer for their positive comments, which we believe will improve the clarity of the manuscript. We address each comment inline below (author response shown in blue).**

**Reviewer:** In Section 2.5.1 I would explicitly state why you are using this complexity metric (i.e. that it links both the model complexity and number of parameters but also the information content of data), instead of the other model complexity metrics that are available. This is stated multiple times elsewhere, but I think it would be useful here as well.

**We agree with the reviewer and have added text to Section 2.5.2 to reflect this suggestion: "The effective complexity of each model run links model structure (i.e., process representation) and number of parameters to the information content of assimilated data. It was computed using a principal component analysis (PCA) on the posterior parameter space."**

**Reviewer:** I'm not 100% sure what point I'm trying to make with this comment so bear with me, but I found myself wanting to dig into more of the nuances of Figure 4, especially in relation the different processes that are included in the model and the level of detail for each process. You do

explore more about the of the differences related to the type of data that are included when you talk about Fig 6, and about the differences across sites with Fig 7. So I found myself wondering about the impact process representation (e.g. ACM v1 vs v2). But I appreciate that's beyond the scope of this study. I guess it just may be worth pointing out (the relatively obvious point) that this framework could also be used to determine which exact model representation is most useful for representing a given dataset – e.g. whether to include a water cycle, or not etc.

**We thank the reviewer for bringing up this point. This is also of interest to us and we are planning to investigate this in future publications. Nonetheless, we have added text in the Discussion (Section 4.3) to address the reviewer's suggestion: "Finally, while beyond the scope of this study, future work will investigate the linkage between specific processes or process representations (e.g., the inclusion or exclusion of water cycling) and predictive performance to better parse ecological controls on the complexity–skill relationship."**

*Minor comments*

**Reviewer:** It would be good to add the prior to Fig. 3, just to see how well the DA system is doing.

**We have added a supplementary figure showing what the reviewer suggests. The supplementary figure, which is also included on the next page, compares NEE predictions produced using model parameters drawn from their prior distributions (blue) to NEE observations (red). For ease of comparison, this figure's panels directly align with the model–site combinations shown in Figure 3.**

**Reviewer:** Unless I am misunderstanding the histogram intersection metric you have described in 2.5.1 would not range between 0 and 1. Perhaps you mean the normalized intersection metric?

**The reviewer is correct—the intersection metric is normalized. We thank them for noting that this was unclear in the text. We have added clarification in Section 2.5.1: "In our case, $p$ was the histogram of predicted NEE or LAI ensembles for a given timestep and $q$ was a discretized Gaussian distribution with mean and standard deviation equivalent to the observed NEE or LAI value and its error, respectively. We normalize the metric by $\sum_{i=1}^{n} p_i$ so that it is bounded between 0 (no overlap) and 1 (identical distributions)."**

[Figure]

**Fig. S1** *(to be included in revised manuscript)*: **Example model runs parameterized strictly using prior distributions at the FR-LBr site. For comparison, panels correspond directly to the models shown in Fig. 3. The calibration window—the first 5 years of the record—is shown in white and the forecast window is shaded gray. The ensemble spread (blue shading) encapsulates the 5th-95th percentile of runs.**

---

## Author Comment (AC2) · 24 Feb 2021

**Author response to interactive comments on "Optimal model complexity for terrestrial carbon cycle prediction"**

**Reviewer:** Famiglietti and colleagues explored relationship between model complexity and forecast skill either with or without assimilated data using a data assimilation system. The authors found that without assimilated data, a complex model has a poorer forecast skill than a simple model; with assimilated data, the opposite is true. The findings make sense and highlight the importance of using data to inform model before forecasting. The manuscript is very interesting and well written. I have only a few minor concerns about the manuscript below.

We thank the reviewer for their feedback. We have responded to each minor concern below (author response shown in blue).

**Reviewer:** L230-232: Will there be any difference in key results and conclusion obtained between using the histogram interaction and using the more familiar metrics?

Our key result—the decline in performance attributable to the most complex models under extreme parametric uncertainty scenarios—is preserved across metrics, as shown on the next page (Figure 1) for the normalized root-mean-square error (RMSE; note that larger values correspond to poorer performance) and coefficient of determination ($R^2$).

We selected the histogram intersection for use in the manuscript because it accounts for prediction accuracy along with prediction and observational uncertainties. Note that because the RMSE and $R^2$ metrics account for only the first, they are less sensitive to the effects of the different factorial combinations (Table 3, main text) on model parameterization across the effective complexity axis and are therefore less interpretable. We chose not to report these results in the manuscript because they only provide an assessment of individual model skill, and do not provide an integrated assessment of both prediction accuracy and uncertainty.

For clarity, however, we have added the following text to section 3.2: "The decline in performance attributable to the most extreme effective complexity scenarios is also preserved across RMSE and $R^2$ metrics (not shown; further comparison between different metrics is beyond the scope of this paper)."

[Figure]

**Figure 1: Comparison of model performance across effective complexity axis for RMSE (top row) and $R^2$ (bottom row) metrics. Left column (a, c) shows all runs included in the experiment; right column (b, d) shows only the subset of runs for which data were assimilated.**

**Reviewer:** L233-239: n value (number of bins) used is?

**We have specified the number of bins (n = 50) in the indicated lines.**

**Reviewer:** L405: "assimilate diverse data types" operates blindly. Some datasets are more useful to constrain a specific variable than other datasets. We can do better than just "diverse".

**We agree with the reviewer and have amended the indicated line as follows: "assimilate *well-characterized, repeat-observation* datasets".**

**Reviewer:** Table 1. Explain the meanings of the IDs (e.g., C groups, S groups). Why the sub-models ordered in the current way in the Table?

Models are ordered in Table 1 alphabetically by model ID. Models are grouped according to common characteristics, as follows: C models all share the Combined Deciduous Evergreen Analytical (CDEA or CDEA+) phenology sub-model; G models use the Growing Season Index (GSI) phenology sub-model; E models use the evergreen (constant allocation) phenology sub-model; and S models are simple, reduced-complexity variants of other models.

We have added this description to the caption of Table 1.

**Reviewer:** Fig. 5a: "(a) All runs", do you mean all runs without assimilating data? You may make it clearer.

"All runs" refers to all runs included in the experiment (that is, both with and without assimilating data). We have amended the subplot's title to read "All runs in the experiment" and the figure caption to read "(a) all model runs in the experiment and (b) the subset of runs in panel (a) for which data were assimilated."

**Reviewer:** Fig. 6: arrangement of the panels are not in a good logic to me. Probably as (a) None ... (f) NEE, LAI, biomass.

The reviewer brings up a worthwhile point. We have rearranged the subplots in Figure 6 so that they are ordered from strongest (NEE, LAI, biomass) to weakest (None) assimilated data constraint, which aligns with our interpretation in the text. The figure caption has also been amended to include the line "Ordering of subplots reflects strongest (a) to weakest (f) data constraint." For consistency, we have made the same changes to Figure S6.

**Reviewer:** Description of skill metric and complexity metric as well as the model structure are clear, but deposit the code to produce this manuscript could be more helpful to others to use the approaches here.

We agree with the reviewer and have made our analysis code publicly available via Github. We have included this statement and URL (github.com/cfamigli/COMPLEX) in the acknowledgments section of the revised manuscript.